# A Whole-Body Physiologically Based Pharmacokinetic Model Characterizing Interplay of OCTs and MATEs in Intestine, Liver and Kidney to Predict Drug-Drug Interactions of Metformin with Perpetrators

**DOI:** 10.3390/pharmaceutics13050698

**Published:** 2021-05-11

**Authors:** Yiting Yang, Zexin Zhang, Ping Li, Weimin Kong, Xiaodong Liu, Li Liu

**Affiliations:** Center of Drug Metabolism and Pharmacokinetics, China Pharmaceutical University, Nanjing 210009, China; 1821010209@stu.cpu.edu.cn (Y.Y.); 1821010211@stu.cpu.edu.cn (Z.Z.); 3119010071@stu.cpu.edu.cn (P.L.); 1831010057@stu.cpu.edu.cn (W.K.)

**Keywords:** drug–drug interaction (DDI), metformin, organic cation transporters (OCTs), physiologically based pharmacokinetic model (PBPK), multidrug/toxin extrusions (MATEs), pharmacokinetics

## Abstract

Transmembrane transport of metformin is highly controlled by transporters including organic cation transporters (OCTs), plasma membrane monoamine transporter (PMAT), and multidrug/toxin extrusions (MATEs). Hepatic OCT1, intestinal OCT3, renal OCT2 on tubule basolateral membrane, and MATE1/2-K on tubule apical membrane coordinately work to control metformin disposition. Drug–drug interactions (DDIs) of metformin occur when co-administrated with perpetrators via inhibiting OCTs or MATEs. We aimed to develop a whole-body physiologically based pharmacokinetic (PBPK) model characterizing interplay of OCTs and MATEs in the intestine, liver, and kidney to predict metformin DDIs with cimetidine, pyrimethamine, trimethoprim, ondansetron, rabeprazole, and verapamil. Simulations showed that co-administration of perpetrators increased plasma exposures to metformin, which were consistent with clinic observations. Sensitivity analysis demonstrated that contributions of the tested factors to metformin DDI with cimetidine are gastrointestinal transit rate > inhibition of renal OCT2 ≈ inhibition of renal MATEs > inhibition of intestinal OCT3 > intestinal pH > inhibition of hepatic OCT1. Individual contributions of transporters to metformin disposition are renal OCT2 ≈ renal MATEs > intestinal OCT3 > hepatic OCT1 > intestinal PMAT. In conclusion, DDIs of metformin with perpetrators are attributed to integrated effects of inhibitions of these transporters.

## 1. Introduction

Metformin is widely used for the treatment of type 2 diabetes. Under physiological pH, metformin is highly ionized, and its transmembrane transport is mainly mediated by transporters, which mainly include organic cation transporter 1–3 (OCT1-3), multidrug/toxin extrusions (MATE1 and MATE2-K) and plasma membrane monoamine transporter (PMAT) [1]. These transporters are distributed in different organs, such as the intestine, liver, and kidney, indicating that disposition of metformin should be attributed to the integrated effect of OCT1-3, MATEs and PMAT in intestine, liver and kidney. OCT1 and OCT2 are mainly expressed on basolateral membrane of enterocytes and show low expression levels in the human enterocytes [2,3,4], inferring a limited role in intestinal absorption of metformin. OCT3 shows a relatively higher expression level on the apical membrane of human enterocytes [3,5]. Several mouse experiments also have demonstrated that silencing OCT3 remarkably impairs intestinal absorption of metformin, whose bioavailability was reported to significantly decreased by 30–53% of wild type mice [6,7]. A single nucleotide polymorphism (SNP) rs12194182 (C > T) of SLC22A3 was reported to be linked to lower mean HbA1c levels in Jordanian type 2 diabetic patients treated with metformin. The subjects with the CC genotype of rs12194182 exhibit the lowest mean HbA1c levels, while patients with the CT and TT genotypes possess higher HbA1c levels [8]. The mean reduction in HbA1c levels following 3-month metformin treatment is higher in Iranian patients with the A allele of rs2292334 (G > A) than in those with the homozygous G allele [9]. Pharmacokinetic analysis showed that Jordanian volunteers with a SLC22A3 rs8187722 variant have higher metformin C_max_ and AUC values than the wild SLC22A3 volunteers. Similarly, volunteers with the heterozygote SLC22A3 rs2292334 variant also have significantly higher metformin C_max_ and AUC values than the wild-type SLC22A3 genotype [10]. These results demonstrate roles of intestinal OCT3 in intestinal absorption of metformin. The absorbed metformin is principally eliminated as unchanged drug via urine. Renal clearance of metformin is 510 mL/min, which is 4.3 folds of glomerular filtration rate (120 mL/min) [11], indicating that renal clearance of metformin is mainly attributed to active secretion via renal tubules. In kidney, metformin is mainly taken into tubule epithelial cells by OCT2 expressed on the basolateral membrane of tubules, then is pumped out of cells to urine via MATE1/2-K expressed on the apical membrane of tubules [1,12]. OCT1 expressed on the sinusoidal membrane of hepatocytes mediates hepatic uptake of metformin [12]. MATE1 expressed on the canalicular membrane of hepatocytes may mediate biliary secretion, but its biliary clearance is negligible.

Metformin is commonly co-administrated with other drugs. Potential drug-drug interactions (DDIs) of metformin with inhibitors of these transporters have been demonstrated when co-administrated with some drugs including cimetidine, pyrimethamine, trimethoprim, dolutegravir, vandetanib, ondansetron, and rabeprazole [13,14,15,16,17,18,19,20]. These perpetrators are inhibitors of OCTs [12,17,18,19,20]. Cimetidine, pyrimethamine, trimethoprim, and ondansetron are also strong inhibitors of MATE transporters [12]. Cimetidine and trimethoprim are also recommended as clinical inhibitors for MATEs by FDA [21]. Importantly, their inhibitions on MATEs are greatly stronger than those on OCTs. For example, cimetidine Ki values for MATE1 and MATE2-K are about 188 and 76 times lower than those for OCT2 inhibition, respectively. Similarly, pyrimethamine K_i_ values for MATE1 and MATE2-K are 55 and 81 folds lower than those for OCT2 inhibition, respectively. Trimethoprim K_i_ values for MATE1 and MATE2-K are 52 and 391 folds lower than those for OCT2 inhibition [12]. These results indicate that DDIs of metformin induced by cimetidine, pyrimethamine, trimethoprim and ondansetron may be mainly attributed to inhibitions of MATEs.

DDIs of metformin with perpetrators are of clinical concern as elevated plasma concentrations of metformin, which are often associated with an increased risk of lactic acidosis (from Glucophage^®^ label). It’s necessary to assess DDIs of metformin with perpetrators to avoid serious clinical consequences. The physiologically based pharmacokinetic (PBPK) model is considered to a powerful tool to explore and quantitatively predict the pharmacokinetics of drugs and the magnitude of DDIs. It is widely applied at increasingly early stages during drug development and is recommended by the US Food and Drug Administration [22] and the European Medicines Agency [23] for the design of clinical DDI trials and population pharmacokinetic studies. Several investigators have successfully developed PBPK model to illustrate transporter-mediated DDI of metformin with cimetidine [24,25,26]. Several investigators have successfully developed a PBPK model to illustrate transporter-mediated DDI of metformin with cimetidine [24,25,26]. However, these studies have focused on transporter-mediated renal secretion without considering intestinal absorption and hepatic disposition of metformin, which does not explain why some drugs (such as verapamil, trimethoprim and rabeprazole) increase plasma exposure to metformin, but little affect or even attenuate antihyperglycemic activity of metformin [15,27]. Roles of intestinal OCT3 and PMAT in intestinal absorption of metformin have been demonstrated [3,5,6,7,28,29]. Inhibition of intestinal OCTs by these perpetrators is also possibly attributed to low concentration of metformin following oral co-administration. Functions of intestinal OCT3 are dependent on pH values. Moreover, the expressions of OCT3 protein and pH values in intestine are regional [30,31]. These indicate that the perpetrators (cimetidine and rabeprazole) may also induce DDI with metformin via affecting intestinal pH values.

The aim of the study was to develop a whole-body PBPK model characterizing interplay of OCTs and MATE1/2-K in intestine, liver and kidney to predict DDIs of metformin with six perpetrators including cimetidine, pyrimethamine, trimethoprim, ondansetron, rabeprazole and verapamil. The Michaelis–Menten (M-M) model was used to illustrate non-linear intestinal absorption of metformin. The concentrations of metformin in liver were simultaneously simulated. The predicted plasma concentration profiles, peak concentration (C_max_) and area under curve (AUC) of metformin were compared with clinic reports. Individual contributions of liver OCT1, intestinal OCT3 (as well as PMAT), renal OCT2 and renal MATEs to metformin disposition and their integrated effects were investigated, respectively. Gastrointestinal transit rate, intestinal pH values and Ki values for OCT2, OCT3 and MATEs were selected for sensitivity analysis.

## 2. Materials and Methods

### 2.1. Collection of Data

DDIs of metformin with perpetrators were collected from publications on PubMed based on the following criterions. (1) Data may come from healthy subjects following single dose or multidose of metformin when co-administrated with perpetrators; (2) metformin and perpetrators were orally administrated to healthy subjects in immediate release formulation; (3) pharmacokinetic profiles or pharmacokinetic parameters such as C_max_ and AUC are shown; (4) DDI data may come from different reports; (5) perpetrators are inhibitors of OCTs or MATEs, whose Ki or IC_50_ values for these transporters are shown.

Model parameters for illustrating pharmacokinetics of metformin and perpetrators as well as DDIs of metformin with perpetrators used in the PBPK model were selected according to following criterions. (1) The optimal parameters of metformin, cimetidine, ondansetron, trimethoprim and verapamil used in the PBPK model were previously reported [24,25,26,32,33,34]; (2) absorption parameter of metformin was derived from data in Caco-2 cells; (3) the reported K_i_ values of some perpetrators often showed large variations. In order to fully investigate risks of DDIs, the smallest K_i_ values (strongest inhibition) were selected. The selected model parameters are listed in Table 1.

### 2.2. Development of PBPK Model

A whole-body PBPK model involving interplay of OCTs and MATEs in intestine, liver and kidney (Figure 1) was developed. The formulas for building the whole-body PBPK model are shown as follows.

In the gastrointestinal tract:

The gut lumen and gut wall (enterocytes) are divided into duodenum, jejunum, ileum, caecum and colon according to their physiological and anatomical characteristics. It is assumed that absorption and metabolism of drug only occur in duodenum, jejunum and ileum. M-M model is used to illustrate non-linear intestinal absorption of metformin. Contribution of intestinal OCT3 and intestinal PMAT to apical to basolateral side of metformin were assumed to be 50% and 20% based on previous reports [6,7] and report [28], respectively. The rested 30% is due to other transporters. Thus, the drug amount in stomach (A_0_), drug amount in intestinal lumen (A_i_, i = duodenum, jejunum and ileum), and in the enterocytes (A_gwi_) are illustrated as follows.
(1)$${\mathrm{dA}}_{0}{/\mathrm{dt}=-K}_{t0}{\;\times \;A}_{0}$$
(2)$$\begin{array}{l} {\mathrm{dA}}_{i}{/\mathrm{dt}\;=\;K}_{\mathrm{ti}-1}{\times A}_{i-1}{\;-\;K}_{\mathrm{ti}}{\times A}_{i}{\;-\;30\%\;\times \;k}_{a,i}{\times \;A}_{i}{\;-\;\mathrm{PBSF}}_{i}\;\times \;\\ {(V}_{\max\_\mathrm{OCT3}}{\;\times \;A}_{\mathrm{gwi}}{/V}_{\mathrm{gwi}}{/K}_{g:b})/\left(K_{m\_\mathrm{OCT3}}{\;+\;A}_{\mathrm{gwi}}{/V}_{\mathrm{gwi}}{/K}_{g:b}\right){\;\times \;T}_{\mathrm{sf},\mathrm{ex},i}{\;\times \;T}_{\mathrm{sf},\mathrm{pH},i\_\mathrm{OCT3}\;-\;}\\ {\mathrm{PBSF}}_{i}{\;\times \;(V}_{\max\_\mathrm{PMAT}}{\;\times \;A}_{\mathrm{gwi}}{/V}_{\mathrm{gwi}}{/K}_{g:b}{)/(K}_{m\_\mathrm{PMAT}}{\;+\;A}_{\mathrm{gwi}}{/V}_{\mathrm{gwi}}{/K}_{g:b}{)\;\times \;T}_{\mathrm{sf},\mathrm{pH},i\_\mathrm{PMAT}} \end{array}$$
(3)$$\begin{array}{l} {\mathrm{dA}}_{\mathrm{gwi}}{/\mathrm{dt}\;=\;Q}_{\mathrm{gwi}}{\;\times \;A}_{\mathrm{art}}/V_{\mathrm{art}}{\;+\;30\%\;\times \;k}_{a,i}{\times A}_{i}{\;+\;\mathrm{PBSF}}_{i}{\;\times \;(V}_{\max\_\mathrm{OCT3}}{\times \;A}_{\mathrm{gwi}}{/V}_{\mathrm{gwi}}{/K}_{g:b})\\ /\left(K_{m\_\mathrm{OCT3}}{+\;A}_{\mathrm{gwi}}{/V}_{\mathrm{gwi}}{/K}_{g:b}\right){\;\times \;T}_{\mathrm{sf},\mathrm{ex},i}{\;\times \;T}_{\mathrm{sf},\mathrm{pH},i\_\mathrm{OCT3}}\\ +{\mathrm{PBSF}}_{i}{\;\times \;V}_{\max\_\mathrm{PMAT}\;}{\times \;A}_{\mathrm{gwi}}{/V}_{\mathrm{gwi}}{/K}_{g:b}\\ /\left(K_{m\_\mathrm{PMAT}}{+\;A}_{\mathrm{gwi}}{/V}_{\mathrm{gwi}}{/K}_{g:b}\right){\;\times \;T}_{\mathrm{sf},\mathrm{pH},i\_\mathrm{PMAT}}\;{-\;Q}_{\mathrm{gwi}}{\;\times \;A}_{\mathrm{gwi}}{/V}_{\mathrm{gwi}}{/K}_{g:b} \end{array}$$
where K_t0_ is constants of gastric emptying rate for stomach. K_ti_ and k_a,i_ are constants of gastric emptying rate and drug absorption rate. K_g:b_ is ratio of drug concentration in gut to blood. The k_a,i_ values were cited from literatures or estimated using human intestinal effective permeability from apical to basolateral side (P_eff,man,A-B_) through equation k_a,i_ = 2$\;{\times \;P}_{\mathrm{eff},A-B}{/r}_{i\;}$. r_i_ values are intestinal radius of duodenum, jejunum and ileum, which are 2.0, 1.63 and 1.45 cm [50], respectively. Q_gwi_ and V_gwi_ are blood flow and volume of the corresponding enterocytes, respectively. A_art_ and V_art_ are amount of drug in artery blood and volume of artery blood, respectively. V_max_ and K_m_ are the maximum velocity and Michaelis–Menten constant for metformin. The values of K_m_ and V_max_ for metformin absorption by OCT3 and PMAT were reported to be 2.46 mM, 12.08 nmol/mg/min and 1.68 mM, 15.28 nmol/mg/min, respectively [51,52]. PBSFs are total amounts of intestinal S9 protein in duodenum, jejunum and ileum, which were calculated to be 2790.65, 14,465.81, and 11,219.5 mg protein based on the previous reports [53,54]. The expression of intestinal OCT3 is regional [31], thus, transporter-mediated parameters in the ith gut segment is also corrected by a relative transporter scaling factor (T_sf,ex,i_). The expressions of OCT3 in jejunum was assumed to be 1. According to previous report [31], the T_sf,ex,i_ values of OCT3 in duodenum, jejunum and ileum were calculated to be 1.19:1:1.155. Function of intestinal OCT3 is dependent on pH. According to a previous report [30], pH values in duodenum, jejunum and ileum were set to be 5.5, 6.5 and 7.5, respectively. Based on reports on cells [29,51], T_sf,pH,i_ was defined as relative values of OCT3 and PMAT function with different pH. The estimated T_sf,pH,i_ values in duodenum, jejunum and ileum for OCT3 and PMAT were 0.35, 0.74, 1 and 1, 0.48, 0.11, respectively.

In the presences of perpetrators, V_max_ for OCT3 ($V_{\max}^{I}$) in the intestine, it is rewritten as:(4)$$V_{\max}^{I}{=V}_{\max}/\left({1+A}_{i}^{I}{/V}_{i}{/K}_{i,\;\mathrm{OCT}3}\right)$$
where $A_{i}^{I}$ and V_i_ are amount of perpetrator in the ith intestine lumen and volume of the ith intestinal lumen. Superscript “I” indicates perpetrators. The V_i_ values of duodenum, jejunum and ileum were estimated to 314.2, 2170.19 and 2609.05 mL based on their length and radius [50,55]. K_i, OCT3_ is inhibition constant of perpetrator on OCT3.

For perpetrators in intestinal lumen and in the enterocytes
(5)$${\mathrm{dA}}_{i}{/\mathrm{dt}=K}_{\mathrm{ti}-1}{\times A}_{i-1}{\;-\;K}_{\mathrm{ti}}{\times A}_{i}{\;-\;k}_{a,i}{\times A}_{i}$$
(6)$${\mathrm{dA}}_{\mathrm{gwi}}{/\mathrm{dt}=Q}_{\mathrm{gwi}}{\times A}_{\mathrm{art}}/V_{\mathrm{art}}{+k}_{a,i}{\times A}_{i}{\;-\;Q}_{\mathrm{gwi}}{\;\times \;A}_{\mathrm{gwi}}{/V}_{\mathrm{gwi}}{/K}_{g:b\;}$$

In the liver compartment:

For metformin and cimetidine, the liver compartment is divided into hepatic blood and hepatocytes. Hepatic uptake of drugs is mainly controlled by OCT1.

In hepatic blood (A_h,b_),
(7)$$\begin{array}{l} {\mathrm{dA}}_{h,b}{/\mathrm{dt}\;=\;(Q}_{h}{\;\times \;A}_{\mathrm{art}})/V_{\mathrm{art}}\\ +\sum {(Q}_{\mathrm{gwi}}{\;\times \;A}_{\mathrm{gwi}}\\ /V_{\mathrm{gwi}}{\;\times \;K}_{g:b}){\;+\;(Q}_{\mathrm{sp}}{\;\times \;A}_{\mathrm{sp}}{)/(V}_{\mathrm{sp}}{\;\times \;K}_{\mathrm{sp}:b})\;-\;(\left(Q_{h}{+Q}_{\mathrm{sp}}+\sum Q_{\mathrm{gwi}}{)\;\times \;A}_{h,b}\right)\\ /V_{h,b}\;-\;\left({\mathrm{CL}}_{\mathrm{int},\mathrm{up},\mathrm{OCT1}}/\left({1+\;(f}_{\mathrm{ub}}^{I}{\;\times \;A}_{h,b}^{I}\right)/{(V}_{h,b}{\times K}_{i,\mathrm{OCT1}})\;)\right){\times \;f}_{\mathrm{ub}}{\times A}_{h,b}\\ /V_{h,b}{\;+\;f}_{\mathrm{ub}}{\times \mathrm{CL}}_{\mathrm{int},\mathrm{pd}}{\times (A}_{h,c}/\left(V_{h,c}{\times K}_{h:b}\right)\;-\;A_{h,b}/V_{h,c}) \end{array}$$

In hepatocytes (A_h,c_):(8)$$\begin{array}{ll} {\mathrm{dA}}_{h,c}/\mathrm{dt}\;=\; & {\mathrm{CL}}_{\mathrm{int},\mathrm{up},\mathrm{OCT1}}/\left({1+(\;f}_{\mathrm{ub}}^{I}\times {\;A}_{h,b}^{I}{)/(V}_{h,b}{\;\times \;K}_{i,\mathrm{OCT1}})\right){\times \;f}_{\mathrm{ub}}{\;\times \;A}_{h,b}\\ & /V_{h,b}{\;-\;f}_{\mathrm{ub}}{\;\times \;\mathrm{CL}}_{\mathrm{int},\mathrm{pd}}{\;\times \;(A}_{h,c}/\left(V_{h,c}{\;\times \;K}_{h:b}\right)-A_{h,b}/V_{h,b})\\ & -f_{\mathrm{ub}}{\times \mathrm{CL}}_{\mathrm{int},\mathrm{met}}{\;\times \;A}_{h,c}/\left(V_{h,c}{\times K}_{h:b}\right) \end{array}$$
where V_h,b_ and V_h,c_ are volume of hepatic blood and hepatocytes, which were assumed to be 608.3 mL and 1081.7 mL, respectively. K_h:b_ and Q_h_ are ratio of drug concentration in liver to blood and blood flow in hepatic artery, separately. CL_int,up,OCT1_, CL_int,pd_ and CL_int,met_ are OCT1-mediated intrinsic clearance of uptake, efflux clearance from hepatocytes to blood and metabolic clearance, respectively. f_ub_ is unbound fraction of drug in blood. K_i, OCT1_ is inhibition constants of perpetrators on OCT1-mediated uptake. Subscript “sp” indicates spleen.

CL_int,met_ in liver may be back-calculated from observed total liver clearance (CL_liver_) using the equations.
(9)$${\mathrm{CL}}_{\mathrm{liver}}{=\mathrm{CL}}_{\mathrm{system}}{\;-\;\mathrm{CL}}_{\mathrm{renal}}$$
(10)$${\mathrm{CL}}_{\mathrm{liver}}={(f}_{\mathrm{ub}}{\times \mathrm{CL}}_{\mathrm{int},\mathrm{met}}{\;\times \;Q}_{L})/\left(f_{\mathrm{ub}}{\times \mathrm{CL}}_{\mathrm{int},\mathrm{met}}{+Q}_{L}\right)$$
where CL_system_ and CL_renal_ are system clearance and renal clearance. Q_L_ is total hepatic blood flow, which equaled to Q_sp_ + Q_h_ + Q_gwi_.

The amount of other perpetrators in liver (A_h_) is illustrated by a well-stirred model, i.e.,
(11)$${\mathrm{dA}}_{h}{/\mathrm{dt}=(Q}_{h}{\times A}_{\mathrm{art}})/V_{\mathrm{art}}+\sum \left(Q_{\mathrm{gwi}}{\times A}_{\mathrm{gwi}}\right)/\;\left(V_{\mathrm{gwi}}{\times K}_{g:b}\right){+(Q}_{\mathrm{sp}}{\times A}_{\mathrm{sp}}{)/(V}_{\mathrm{sp}}{\times K}_{\mathrm{sp}:b}{)\;-\;(Q}_{h}{\times A}_{h}{)/V}_{h}{\;-\;f}_{\mathrm{ub}}{\times \mathrm{CL}}_{\mathrm{int},\mathrm{met}}{\times A}_{h}/\left(V_{h}{\times K}_{h:b}\right)$$

In the renal compartment:

For metformin and cimetidine, renal excretion of drug mainly occurs via glomerular filtration and renal secretion. It was assumed that reabsorption does not occur. Renal compartment consists of blood compartment and tubule. The renal secrete clearance (CL_ren_) of drug is controlled by OCT2 and MATE1/MATE2-K in series.

Amounts of drugs in renal blood compartment (A_r,b_) and in tubule(A_r,e_) are illustrated as follows.
(12)$$\begin{array}{l} {\mathrm{dA}}_{r,b}{/\mathrm{dt}\;=\;(Q}_{r}{\times A}_{\mathrm{art}})/V_{\mathrm{art}}{\;-\;(Q}_{r}{\times A}_{r,b})\\ /V_{r,b}{\;-\;(\mathrm{RAF}\times \mathrm{CL}}_{\mathrm{int},\mathrm{up},\mathrm{OCT2}}{)/(1+(f}_{\mathrm{ub}}^{I}{\;\times \;A}_{r,b}^{I}{)/(V}_{r,b}{\times K}_{i,\mathrm{OCT2}}{))\times \;f}_{\mathrm{ub}}{\times A}_{r,b}\\ /V_{r,b}{+\;\mathrm{CL}}_{\mathrm{int},\mathrm{up},\mathrm{OCT2}}/\left({1+\;f}_{\mathrm{ub}}^{I}{\;\times \;A}_{r,e}^{I}{/(V}_{r,e}{\times K}_{i,\mathrm{OCT2}})\right){\times \;f}_{\mathrm{ub}}{\times \;A}_{r,e}\\ /\left(V_{r,e}{\times K}_{r:b}\right){-\;f}_{\mathrm{ub}}{\times \mathrm{GFR}\times A}_{r,b}/V_{r,b} \end{array}$$
(13)$$\begin{array}{l} {\mathrm{dA}}_{r,e}{/\mathrm{dt}=(\mathrm{RAF}\times \mathrm{CL}}_{\mathrm{int},\mathrm{up},\mathrm{OCT2}})/\left({1+(f}_{\mathrm{ub}}^{I}{\;\times \;A}_{r,b}^{I}{)/(V}_{r,b}{\times K}_{i,\mathrm{OCT2}})\right){\times \;f}_{\mathrm{ub}}{\times A}_{r,b}\\ /V_{r,b}{-\;\mathrm{CL}}_{\mathrm{int},\mathrm{up},\mathrm{OCT2}}\\ /\left({1+(f}_{\mathrm{ub}}^{I}{\;\times \;A}_{r,e}^{I}{/(V}_{r,e}{\times K}_{i,\mathrm{OCT2}})\right)\times \;\\ f_{\mathrm{ub}}{\times A}_{r,e}{/(V}_{r,e}{\times K}_{r:b}{))\;-\;(\mathrm{RAF}\times \mathrm{CL}}_{\mathrm{int},\mathrm{eff},\mathrm{MATE}}{)/(1+(f}_{\mathrm{ub}}^{I}{\;\times \;A}_{r,e}^{I}{)/(V}_{r,e}{\times K}_{i,\mathrm{MATE}}))\;\times \;\\ f_{\mathrm{ub}}{\times A}_{r,e}/\left(V_{r,e}{\times K}_{r:b}\right) \end{array}$$
where V_r,b_ and V_r,e_ are renal blood volume and tubule volume, respectively. V_r,b_ was reported to be 33.8 mL [56], thus V_r,e_ was estimated to be 246.2 mL. K_r:b_ and Q_r_ are ratio of drug concentration in renal to blood and blood flow in renal, respectively. OCT2-mediated uptake clearance (CL_int,up,OCT2_) and MATE-mediated efflux clearance (CL_int,eff,MATE_) of metformin were reported to be 14.2 μL/min/10^6^ tubules and 16.6 μL/min/10^6^ tubules [25]. Renal uptake of cimetidine is mediated by OAT3 and OCT2. The OAT3-mediated uptake clearance and OCT2-mediated uptake clearance were reported to be 7.63 and 29.89 μL/min/10^6^ tubules [25], respectively. MATE1-mediated efflux clearance and MATE2-K-mediated efflux clearance of cimetidine were reported to be 17.67 and 11.87 μL/min/10^6^ tubules, separately [25]. It was assumed that 60 million proximal tubule cells per gram kidney, and 4.3 g of kidney per kilogram of body weight [57]. GFR is glomerular filtration rate, which was set to be 120 mL/min/70 kg. RAF is empirical scaling factor, being set to be 3.0 [25]. K_i,OCT2_ and K_i,MATE_ is inhibition constants of perpetrators on OCT2-mediated uptake and MATE-mediated efflux.

Well-stirred model was also used to illustrate disposition of other perpetrators in kidney (A_r_), i.e.,
(14)$${\mathrm{dA}}_{r}{/\mathrm{dt}=(Q}_{r}{\times A}_{\mathrm{art}}{)/V}_{\mathrm{art}}-{(Q}_{r}{\times A}_{r}{)/V}_{r}{-\;f}_{\mathrm{ub}}{\;\times \;\mathrm{CL}}_{\mathrm{int},\;\mathrm{renal}}{\;\times \;A}_{r}/\left(V_{r}{\times K}_{r:b}\right)$$

CL_int,renal_ is also back-calculated from observed renal clearance (CL_renal_) using Equation (15).
(15)$${\mathrm{CL}}_{\mathrm{renal}}=\left(f_{\mathrm{ub}}{\times \mathrm{CL}}_{\mathrm{int},\mathrm{renal}}{\times Q}_{r}\right)/\left(f_{\mathrm{ub}}{\times \mathrm{CL}}_{\mathrm{int},\mathrm{renal}}{+Q}_{r}\right)$$

In artery blood (A_rt_) and venous blood (A_ven_).

Amounts of drug in artery blood (A_rt_) and venous blood (A_ven_) are illustrated by
(16)$${\mathrm{dA}}_{\mathrm{art}}{/\mathrm{dt}=Q}_{\mathrm{total}}{\times ((A}_{\mathrm{lu}}{/V}_{\mathrm{lu}}{)/K}_{\mathrm{lu}:b}\;{-\;A}_{\mathrm{art}}{/V}_{\mathrm{art}})$$
(17)$${\mathrm{dA}}_{\mathrm{ven}}/\mathrm{dt}=\sum {\;Q}_{j}{\;\times \;(A}_{j}{/V}_{j}{)/K}_{j:b}{-\;Q}_{\mathrm{total}}{\;\times (A}_{\mathrm{ven}}{/V}_{\mathrm{ven}})$$
where Q_total_, V_lu_, V_ven_, A_lu_ and K_lu:b_ are cardiac output, lung volume, venous volume, amount of drug in lung and ratio of drug concentration in lung to blood, respectively. Subscript “j” indicates other tissues in human body, such as heart, brain, muscle, adipose, skin, and rest tissues.

Phoenix WinNonlin 8.1 (Pharsight, St. Louis, MO, USA) was used for coding and solving of the PBPK model as well as estimating corresponding kinetic parameters (C_max_ and AUC).

### 2.3. Model Validation

Plasma concentration profiles and the plasma exposure parameters (AUC and C_max_) of metformin and 6 perpetrators following oral administration to human were first simulated using the developed PBPK model and the developed PBPK model was validated by visual predictive checks (VPCs). The 5th, 50th, and 95th percentiles of the simulations and their 90% confident intervals were plotted along with the observed data. Following validation, the developed PBPK model was further used to predict DDIs of metformin with perpetrators. The predictions were compared with clinic observations. Extent of DDI was assessed as ratio of AUC (AUCR) or of C_max_ (C_max_R) with perpetrators and without perpetrators. Fold errors, ratios of prediction to observation, were often used to assess prediction. The predictions were considered successful if the ratio of predication to observation fell within 0.5 and 2.0 [58,59]. Both relative squared error (RSE) (Equation (18)) and the geometric mean-fold error (GMFE) (Equation (19)) were further introduced to describe the difference between predictions and observations [60,61].
(18)$$\mathrm{RSE}=\frac{\sum_{i=1}^{n}{{(\mathrm{Pre}}_{i}{-\mathrm{Obs}}_{i})}^{2}}{\sum_{i=1}^{n}{\left(\bar{\mathrm{Obs}}{-\mathrm{Obs}}_{i}\right)}^{2}}\;\times 100\%$$
(19)$${\mathrm{GMFE}=10}^{\frac{1}{n}\sum_{i=1}^{n}\left|\log_{10}(\frac{{\mathrm{Pre}}_{i}}{{\mathrm{Obs}}_{i}})\right|}$$
where Pre_i_, Obs_i_, and $\bar{\mathrm{Obs}}$ represent the predicted parameters, the observed parameters and their average values of observations, respectively. *n* is the number of predictions.

### 2.4. Sensitivity Analysis of Model Parameters

Renal secretion of the metformin is attributed to interplay of renal OCT2 and MATEs. Intestinal absorption of metformin is mediated by intestinal OCT3, PMAT and other transporters. The function of intestinal OCT3 is dependent on pH. Gastrointestinal transit also affects intestinal absorption of metformin. Sensitivity analysis was operated on K_i, OCTs_, K_i, MATE_, intestinal pH and constant of gastrointestinal transit rate.

Metformin is substrates of OCT1, OCT2, OCT3, PMAT, MATE1 and MATE2-K. These transporters are differently expressed in intestine, liver and kidney. The individual contributions of intestinal OCT3 (V_max,OCT3_), intestinal PMAT (V_max,PMAT_), hepatic OCT1 (CL_int,up,OCT1_), renal OCT2 (CL_int,up,OCT2_) and renal MATEs (CL_int,efflux,MATE_) to metformin disposition were investigated.

## 3. Results and Discussion

### 3.1. Collection of DDI Data

Six perpetrators including cimetidine, pyrimethamine, trimethoprim, ondansetron, rabeprazole, and verapamil were selected to investigate DDIs with metformin. Some diseases (such as diabetes and renal failure) altered metformin disposition via affecting physiological parameters of body and expressions/function of drug transporters. For example, diabetes was reported to alter pharmacokinetics of metformin via affecting gastrointestinal transit and renal function [62]. To better investigate the DDI between metformin and perpetrators and to estimate contributions of drug transporters in the liver, kidney, and intestine to metformin pharmacokinetics, the clinical PK data mainly came from healthy volunteers, which come from clinic reports on PubMed. The ratios of drug concentration in tissue to plasma were calculated (Appendix A) using method [63] based on tissue composition and physicochemical parameters of drugs. The physiological parameters (tissue blood flow and volume) (Appendix A) and pharmacokinetic parameters (transporter parameters, metabolism parameters or inhibition parameter K_i_) (Table 1) were cited from corresponding literatures.

Metformin is substrate of OCT1-3 and MATE1/2-K, which are individually expressed in intestine, liver and kidney. Several evidences have demonstrated that ranks for gene expressions of transporters in human intestine [28,64,65] are OCT3 > serotonin transporter (SERT) > PMAT and OCT1. Metformin is a substrate of OCT1, SERT, PMAT and OCT3, whose apparent Km values are 3.1, 4, 1.68, and 2.46 mM, respectively [28,51], indicating that metformin affinities to transporters are PMAT > OCT3 > OCT1 > SERT. Based on expression of transporters and their affinities to metformin, OCT3 and PMAT may be main uptake transporters for metformin in human intestine. SNPs of OCT3 significantly affect intestinal absorption of metformin [66,67], demonstrating important roles of OCT3 in intestinal absorption of metformin, and some reports have showed that PMAT, OCT1 and SERT are involved in metformin transport in Caco-2 cells and other cell lines [11,28,51]. Based on data from OCT3 knockout mice and cell lines, we assumed that contribution of intestinal OCT3 and PMAT to intestinal absorption of metformin was about 50% and 20%, respectively, the rest (30%) was attributed to other transporters including THTR2, OCT1 and maybe SERT [6,7,66,67]. These results indicate that the intestinal absorption of metformin is mediated at least partly by OCT3 and PMAT expressed on apical membrane of enterocytes. Metformin eliminates mainly via renal secretion due to sequential works of OCT2 and MATE1/2-K on basolateral membrane and apical membrane of tubule, respectively. Hepatic uptake of metformin is mediated by liver OCT1.

### 3.2. Quantitatively Predicted Disposition Kinetics for Metformin and Perpetrators

The developed PBPK model considering alliance of OCTs and MATE1/2-K in intestine, liver and kidney was applied to predict plasma concentrations (Figure 2) and their main pharmacokinetic parameters C_max_ and AUC (Table 2) of metformin and 6 perpetrators (cimetidine, pyrimethamine, trimethoprim, ondansetron, rabeprazole and verapamil) using the list parameters (Table 1, Appendix A). The results show that the most of predicted concentrations are within 0.5–2.0 folds of observed data. In line, 94.1% (32/34) of predicted plasma exposure parameters (C_max_ and AUC) also fall within 0.5–2.0 folds of clinic data. RSE for C_max_ and AUC were calculated to be 1.1% and 9.6%, respectively. The estimated GMFE values for C_max_ and AUC were 1.004 and 0.890, near to 1.0. All these results demonstrate that both the developed PBPK model and the selected corresponding parameters are appropriate for describing pharmacokinetics of metformin and perpetrators.

**Table 2 pharmaceutics-13-00698-t002:** The predicted (Pre) and observed (Obs) pharmacokinetic parameters of metformin and perpetrators.

Drug	Ref.	Dose (mg)	C_max_ (μg/mL)		Ratio	AUC_0–t_ (μg·h/mL)		Ratio
			Pre	Obs	Pre/Obs	Pre	Obs	Pre/Obs
Metformin	[68]	750	1.22	1.5	0.81	11.05	9.4	1.18
	[69]	500	0.95	1.55	0.61	7.04	9.08	0.78
Cimetidine	[70]	400	2.06	2.20	0.94	9.17	8.03	1.14
	[71]	300	1.53	1.53	1.00	6.68	5.22	1.28
Pyrimethamine	[72]	50	0.57	0.76	0.75	31.18	76	0.41
	[73]	50	0.57	0.37	1.54	31.18	42.83	0.73
	[74]	75	0.86	0.86	1.00	46.77	124.6	0.38
	[75]	75	0.86	0.60	1.43	46.77	68.34	0.68
Trimethoprim	[76]	210	2.18	2.35	0.93	28.26	37.1	0.76
	[77]	1400	13.76	12.78	1.08	302.34	299.31	1.01
Ondansetron	[78]	8	0.046	0.0272	1.69	0.203	0.198	1.03
	[79]	8	0.046	0.037	1.24	0.233	0.254	0.92
Rabeprazole	[80]	40	0.40	0.502	0.80	1.27	1.315	0.97
	[80]	40	0.40	0.444	0.90	1.27	1.332	0.95
	[81]	20	0.20	0.252	0.79	0.63	0.575	1.10
Verapamil	[82]	40	0.050	0.033	1.52	0.28	0.22	1.27
	[83]	80	0.099	0.13	0.76	0.564	0.387	1.46

**Figure 2 pharmaceutics-13-00698-f002:**
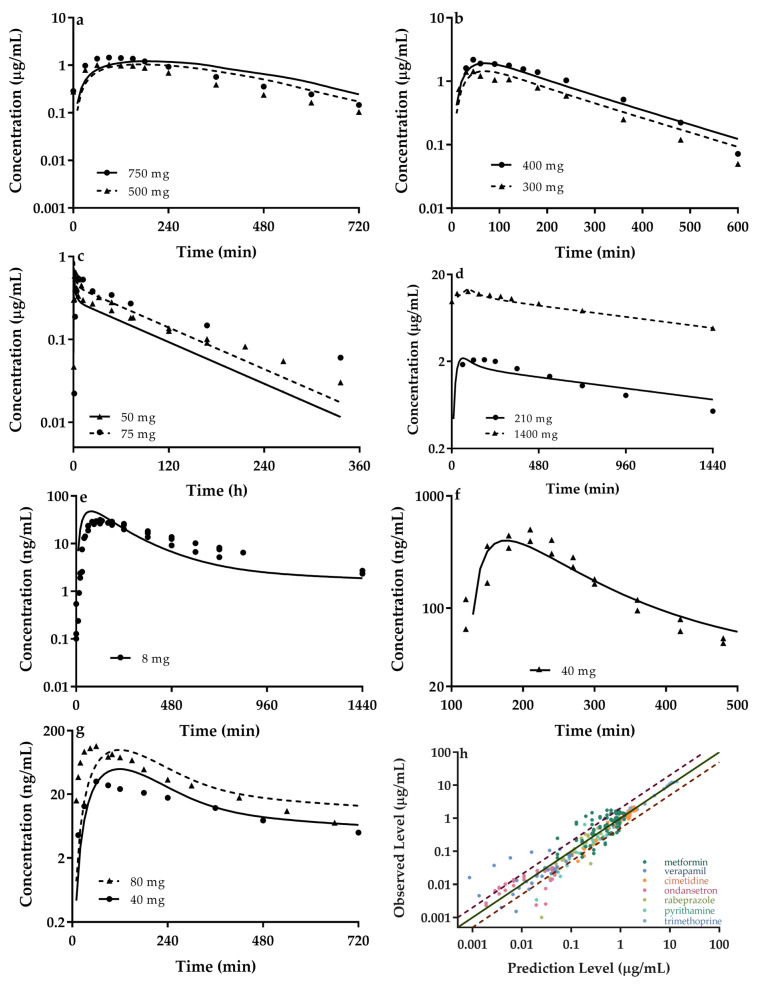
Predicted (lines) and observed (points) plasma concentrations of (**a**) metformin and (**b**) cimetidine, (**c**) pyrimethamine, (**d**) trimethoprim, (**e**) ondansetron, (**f**) rabeprazole (with the lag time because of dosage form) and (**g**) verapamil following oral dose to human. The observations were cited from reports [68,70,71,72,74,76,77,78,79,82,83,84,85,86,87]. (**h**) Relationship between the observed and predicted plasma concentration of the seven tested agents with different colors. Solid and dashed lines respectively represent unity and 2-fold errors between observed and predicted data.

M-M model is successfully used to illustrate OCT3-and PMAT-mediated intestinal absorption of metformin absorption following 50–1000 mg of oral metformin to human (Figure 3a, Appendix A). The results show that absorption kinetics is characterized by non-linear kinetics and that the oral clearance (dose/AUC) is increased along with dose, in line with the observations (Figure 3b), inferring that non-linear pharmacokinetics of metformin mainly results from transporter-mediated intestinal absorption.

Visual predictive checks (VPCs) were made for validating the model in human populations and assessing the accuracy of the predictions following oral single dose (500 mg) administration of metformin to human. Intestinal OCT3 (V_max,OCT3_), liver OCT1 (CL_int,up,OCT1_), renal OCT2 (CL_int,up,OCT2_), renal MATEs (CL_int,efflux,MATEs_), and gastrointestinal transit rate (K_ti_) of metformin were investigated as factors for inter-individual variability. Exponential model and multiplicative residual error model were used to simulate the inter-individual variability and intra-individual variability of these parameters. The parameter estimation method was the first order conditional estimation of Lindstrom-Bates (FOCE L-B) [115]. Together with standard deviation of intra-individual error, V_max,OCT3_, CL_int,up,OCT1_, CL_int,up,OCT2_, CL_int,efflux,MATEs_, and K_ti_, which were regarded as random effect parameters were simulated with six sets of the observation data. Subsequently, visual predictive checks were performed on Phoenix NLME module (version 1.3, Certara, Co., Princeton, NJ, USA) based on 1000 times of simulations. VPCs were indicated as comparison between 5, 50, 95 percentiles of the observations and the corresponding simulations (Figure 3c). The virtual trial simulation showed that 94.4% observations are within the fifth and 95th percentiles of the simulated populations, indicating that good predictions for metformin are achieved by the developed whole-body PBPK model.

### 3.3. Predicted DDIs of Metformin with Perpetrators

The selected perpetrators are all strong inhibitors of OCTs, some of which (cimetidine, pyrimethamine, trimethoprim and ondansetron) are stronger inhibitors of MATE1/2-K [12], indicating that DDIs of metformin with these perpetrators are attributed to the integrated effects of inhibition on intestinal OCT3, renal OCTs, and renal MATE1/2-K. Following validating the developed PBPK model in individual compounds, the PBPK was scaled to predict DDIs of metformin with 6 perpetrators according to the administration schedules in Appendix A. The predictions were compared with clinic data (Figure 4 and Table 3).

It was found that most of predicted concentrations are within 0.5–2.0 folds of observed concentrations. In line, 86.4% (57/66) predicted plasma exposure parameters (C_max_ and AUC) for DDI also fall within 0.5–2.0 folds of observations, inferring that the developed PBPK model may be used to successfully predict the DDIs of metformin and its possible mechanisms.

Among the tested perpetrators, cimetidine, pyrimethamine, trimethoprim and ondansetron show stronger inhibitions on MATE1/2-K than those on OCTs, inferring that the four perpetrators increase plasma exposure to metformin mainly via inhibiting renal secretion mediated by MATE1/2-K. However, intestinal concentrations of perpetrators may be higher than their K_i_ values for OCT3 following oral dose. For example, following oral 400 mg cimetidine, 50 mg pyrimethamine, 200 mg trimethoprim and 8 mg ondansetron, concentration of cimetidine, pyrimethamine, trimethoprim and ondansetron in intestinal lumen were estimated to be 6.34 mM, 0.80 mM, 2.76 mM, and 109 μM using dose/250 mL, respectively, which are greatly higher than K_i_ values for OCT3 (45.7 μM for cimetidine, >100 μM for pyrimethamine, 12.3 μM for trimethoprim and 17.4 μM for ondansetron), indicating that inhibition of intestinal OCT3 also contributes to DDIs of metformin, although the contribution of intestinal OCT3 inhibition is contrast to that of renal OCT2 and MATE1/2-K inhibitions. No evidence demonstrates that verapamil inhibits functions of MATEs, K_i_ values of verapamil for OCT2 and OCT3 are similar. Following oral 180 mg verapamil, concentration of verapamil in intestinal lumen was estimated to be 1.58 mM, which is higher than K_i_ values for OCT3 (3.6 μM). The inhibition of intestinal OCT3 by verapamil partly abolishes the increased plasma exposure to metformin by inhibitions of renal OCT2, which is in line with little alterations in plasma concentrations of metformin. Simulation shows that rabeprazole, a weak inhibitor of OCTs and MATE1, does not affect plasma concentrations of metformin, which is consistent with clinical observations [20].

Liver is a main targeted organ for metformin. Roles of hepatic OCT1 in plasma exposure to metformin is minor, but distribution of metformin in liver is highly controlled by hepatic OCT1, in turn, affecting antihyperglycemic activity of metformin. Concentrations of metformin in liver were simultaneously simulated following coadministration of these perpetrators. The simulation data shows that coadministration of trimethoprim, rabeprazole and verapamil may increase plasma concentrations of metformin, while obviously decreased concentrations of metformin. Although the decreases in concentration of metformin in liver by the three perpetrators needed to be supported by clinical data, the simulations partly explained the curious phenomenon that why co-administration of trimethoprim, rabeprazole and verapamil little affect or attenuate antihyperglycemic activity of metformin in clinical [15,99,103].

### 3.4. Sensitivity Analysis

Metformin mainly eliminates via renal secretion, which is highly controlled by renal OCT2 and MATEs. Intestinal absorption of metformin is mainly mediated by intestinal OCT3 and PMAT. Effects of K_i_ values for OCTs and MATEs, intestinal pH values and gastrointestinal transit rate on DDI of metformin were investigated using cimetidine as a representative for sensitive analysis. The variabilities of these parameters were based on the reality and reports. K_i_ values of cimetidine for OCT2, OCT3, MATE1 and MATE2-K [49] are reported to vary 657 folds, 11 folds, 61 folds, and 22 folds, respectively. Thus, the variabilities of K_i_ values were set to be 0.1, 1 and 10, respectively. Variabilities in gastrointestinal transit time were set to be 0.5, 1, and 2. Moreover, function of intestinal OCT3 and PMAT is dependent on pH values. Based on physiological structural characteristics, three pH conditions were taken into account. Condition 1 (control): pH values in duodenum, jejunum and ileum were set to be 5.5, 6.5, and 7.5, respectively. Condition 2: pH values in duodenum, jejunum and ileum were set to be 6.5, 7.5, and 7.5, respectively. Conditions 3: pH values in duodenum, jejunum and ileum were set to be 7.5, 7.5, and 7.5, respectively. It was found that the tested factors remarkably alter DDIs of metformin with cimetidine (Figure 5). The contribution of gastrointestinal transit rate is the strongest, followed by K_i,MATEs_, K_i,OCT2_, K_i,OCT3_, intestinal pH, and K_i,OCT1_. The decrease in gastrointestinal transit rate by 50% of control increases AUC of metformin by 39.7%. In contrast, 2-fold increase gastrointestinal transit rate leads to decrease in AUC of metformin by 38.6%. In line with a previous report [116], a decrease gastrointestinal transit rates increases plasma exposure to metformin. Effects of cimetidine on plasma exposure to metformin varied with K_i_ values for MATEs/OCTs, which was dependent on their contribution. Alterations in K_i_ values for OCT2 and MATE1 significantly altered AUCR of metformin. For example, AUCR values at 0.1 × K_i_, 1 × K_i_ (0.65 μM) and 10 × K_i_ of MATEs were estimated to be 1.23, 1.54 and 1.98, respectively. OCT3 mainly affected metformin absorption, altering C_max_ of metformin. The estimated C_max_R values at 0.1 × K_i_, 1 × K_i_ (45.7 μM) and 10 × K_i_ of OCT3 were 1.47, 1.28 and 1.03, respectively. Alterations in K_i_ values for OCT1 little affected plasma exposure to metformin, consistent with minor roles of OCT1 in metformin disposition.

Functions of OCT3 and PAMT are also dependent on pH [29,51] and intestinal pH values are also regional [30]. Both cimetidine and rabeprazole themselves inhibit secretion of gastrointestinal acids, leading to increase in intestinal pH. However, the effects of pH on functions of OCTs and PAMT are opposing and the simulation showed that alteration in intestinal PMAT-mediated metformin absorption by pH increase is larger than that of intestinal OCT3. The increased intestinal pH enhances OCT3-mediated intestinal absorption but decrease PMAT-mediated intestinal absorption of metformin, indicating that net contributions of increases in intestinal pH by cimetidine and rabeprazole to intestinal absorption of metformin is minor.

Individual contributions of intestinal OCT3 (V_max,OCT3_) and PMAT (V_max,PMAT_), liver OCT1 (CL_int,up,OCT1_), renal OCT2 (CL_int,up,OCT2_) and MATEs (CL_int,efflux,MATEs_), and the integrated effects (CL_int,up,OCT2_ + CL_int,efflux,MATEs_) to metformin disposition were also investigated (Figure 6). Variabilities of these parameters were set to be 0 and 1. The results demonstrated that both OCT-mediated intestinal absorption and OCT/MATE-mediated renal secretion showed important roles in metformin disposition, although contributions of intestinal OCT3 and PMAT to plasma concentrations of metformin are contrary to those of renal OCT2 and MATEs. Contributions of the individual transporters to AUC of metformin were assessed using (AUC_non transporter_ − AUC_control_)/AUC_control_) × 100%, where AUC_non-transporter_ is AUC of metformin without considering the transporter. The results (Figure 6b) showed that contributions of these transporters to AUC of metformin are renal OCT2 (160.30%) ≈ renal MATE (159.80%) > intestinal OCT3 (−35.94%) > hepatic OCT1 (15.67%) > intestinal PMAT (−12.06%). A decrease in intestinal absorption of metformin due to inhibition of intestinal OCT3 may be partly attenuate increase of plasma exposure to metformin by inhibitions of renal OCT2 and MATE1/2-K.

Interestingly, compared with deleting renal OCT2 or renal MATEs alone, simultaneously deleting renal OCT2 and renal MATE no longer enhanced AUC of metformin, the AUC of metformin was 12.50 μg·h/mL, which was near to those of deleting renal OCT2 (12.50 μg·h/mL) and renal MATEs (12.48 μg·h/mL). The renal secretion of metformin is attributed to sequential work of uptake transporter OCT2 and efflux transporter MATEs (Figure 7), which may partly explain that renal OCTs and MATEs have the same contribution to drug disposition. The simulation showed that the total clearance for both delete renal OCT2 and renal MATEs (166.7 mL/min, assume bioavailability to be 50%) is similar to the GFR (132.9 mL/min).

These simulations have demonstrated importance of OCT1, OCT2, OCT3 and MATE1/2-K in metformin disposition and metformin antihyperglycemic activity. Diabetes was reported to increase plasma exposure to metformin following oral dose, accompanied by decrease in renal and system clearance [117]. Creatinine clearance is also impaired, which highly correlated with renal clearance [117]. Creatinine is excreted into urine by glomerular filtration and renal tubular secretion mediated by organic anion transporter 2 (OAT2), OCT2, OCT3 and MATE1/MATE2-K [49]. Animal experiments have shown that diabetes downregulates expression of renal OCT2 and OCT3 [118,119], partly explaining decreases in renal clearance and system clearance of metformin which may contribute to decrease in creatinine clearance. Moreover, the increase in plasma exposure to metformin under diabetic status is partly attributed to decrease in gastrointestinal transit time [62].

However, the developed PBPK model also has some limitations. Here, we assumed that 70% of intestinal metformin absorption was attributed to intestinal OCT3 and PMAT expressed on apical membrane of enterocytes, and the rested 30% was due to other transporters (such as OCT1, serotonin reuptake transporter and choline high-affinity transporter), whether are suitable also needs further investigation. Moreover, some diseases such as diabetes affected DDIs of metformin with perpetrators via affecting physiological parameters of body and expressions/function of drug transporters, which was reported before [62].

Several different ways (such as optimizing the equations, improving the parameters, and rearranging the model structure involved in absorption, distribution, metabolism and excretion) have been tried to close the gap between prediction and observation, but the discrepancies between prediction and observation still exist (Figure 2 and Figure 4). The discrepancies may come from such as genotype, sex and age. A report showed that gender and race (Caucasians, Blacks, and Hispanics) little affected AUC of metformin [120]. The elderly subjects (68 ± 2 years) were reported to exhibit 1.7 and 2.0 times higher average C_max_ and AUC than the younger subjects (23 ± 3 years) [93]. Subjects used in the prediction were young and healthy, excluding effects of both ages and disease on metformin disposition. Genetic variants of OCTs/MATEs have been demonstrated to alter metformin disposition [10,121,122,123], but results are often confused. For example, the renal clearance of metformin was unaltered in patients carrying the MATE1 variant, OCT2 and OCT3 [121]. However, volunteers with SLC22A3 variants (rs8187722 or rs2292334) had higher C_max_ and AUC of metformin than the wild SLC22A3 [10]. Another report showed that compared with OCT2 reference allele (808G/G), volunteers carrying heterozygous for 808G/T had higher renal clearance and secretory clearance of metformin, accompanied by significantly lower metformin concentrations at early times after metformin administration [122]. However, c.808 (G > T) alone affected neither renal clearance nor secretory clearance of metformin, but both the renal clearance and secretory clearance were significantly increased for the volunteers with c.808 (G > T) who were also homozygous for the reference variant g.−66T > C in MATE1. On the contrast, the volunteers with c.808 (G > T) who were also heterozygous for g.−66T > C showed the lower renal clearance and secretory clearance of metformin compared with volunteers with c.808 (G > T) carrying the g.−66T > C reference genotype. These results indicated that c.808 (G > T) could have a dominant genotype to phenotype correlation [123]. Thus, the contributions of OCT and MATE polymorphisms to pharmacokinetics of metformin and DDI needed further investigation.

## 4. Conclusions

In conclusion, the PBPK model characterizing the interplay of OCTs and MATEs is successfully applied to predict the pharmacokinetics of metformin and its DDIs with perpetrators. Perpetrator-induced DDIs of metformin should be attributed to integrated effects of intestinal OCT3, renal OCT2/MATE1/2-K, and hepatic OCT1 inhibitions.

## Figures and Tables

**Figure 1 pharmaceutics-13-00698-f001:**
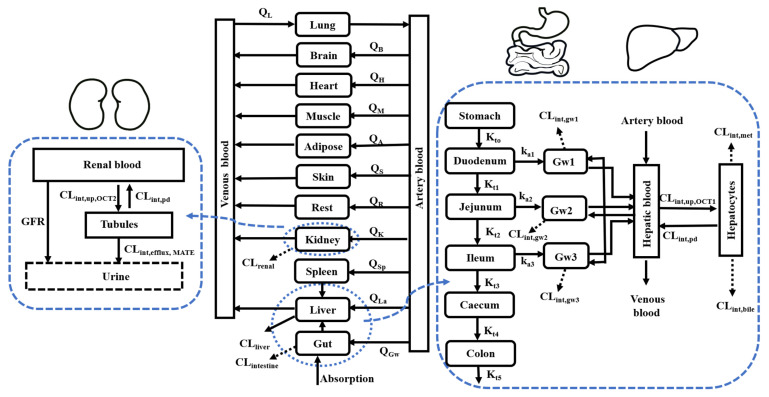
Schematic diagram of PBPK model involving interplay of OCTs and MATEs in intestine, liver and kidney. Qi indicates blood flow in corresponding compartment. CL_int_ is intrinsic clearance. K_ti_ and k_a,i_ represent the transit rate constant and drug absorption rate constant, respectively. CL_int,up_, CL_int,pd_, CL_int,bile_ and CL_int,met_ are OCT1-mediated uptake, efflux clearance to blood, efflux clearance to bile and metabolism clearance in hepatocytes. CL_int,up, OCT2_ and CL_int,efflux,MATE_ are OCT2-mediated uptake clearance and MATE-mediated efflux clearance to urine. GFR is glomerular filtration rate.

**Figure 3 pharmaceutics-13-00698-f003:**
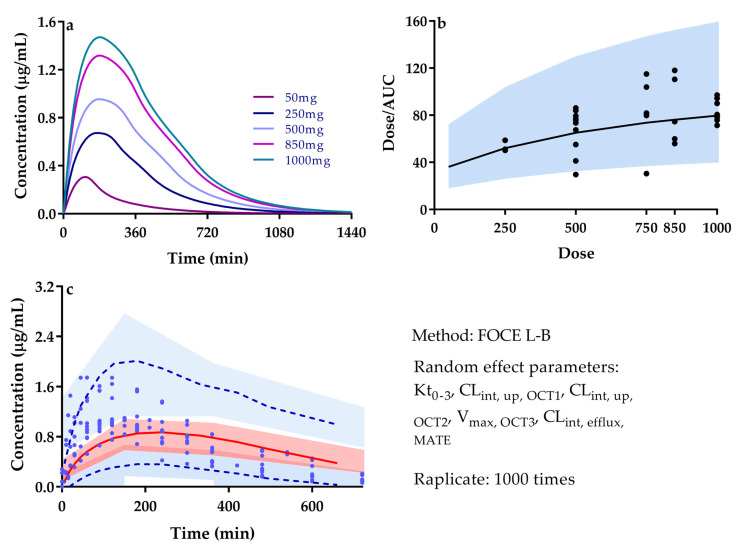
(**a**) Predicted plasma concentrations of metformin following different dosage of human. (**b**) Predicted (line) and observed (point) relationship between metformin dose and dose/AUC for different dose; shaded area, 0.5–2.0 folds of prediction; observations were cited from [13,15,16,19,20,27,68,69,84,88,89,90,91,92,93,94,95,96,97,98,99,100,101,102,103,104,105,106,107,108,109,110,111,112,113]. (**c**) Visual predictive checks (VPCs) of metformin plasma concentrations to time in 500 mg oral administration for human; solid line, the 50th percentiles; dashed lines, the 5 and 95th percentiles of the simulated populations; the shaded area, 90% confidence intervals of the simulated concentrations of the 5, 50 and 95th percentiles; and the point, observations, which were cited from [27,69,91,93,97,114].

**Figure 4 pharmaceutics-13-00698-f004:**
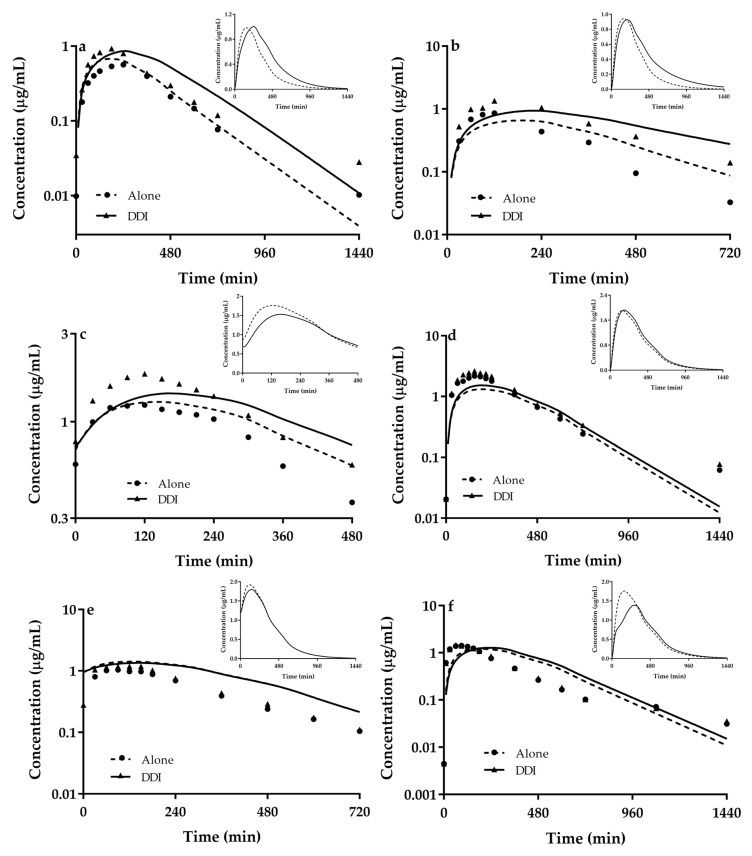
Predicted (line) and observed (points) plasma concentrations of the metformin alone and co-administrated with (**a**) cimetidine, (**b**) pyrimethamine, (**c**) trimethoprim, (**d**) ondansetron, (**e**) rabeprazole and (**f**) verapamil. Insert plots indicated concentrations of metformin in liver. Observations were cited from reports [13,14,16,19,27,99]. Concentration of metformin in liver have no clinical data.

**Figure 5 pharmaceutics-13-00698-f005:**
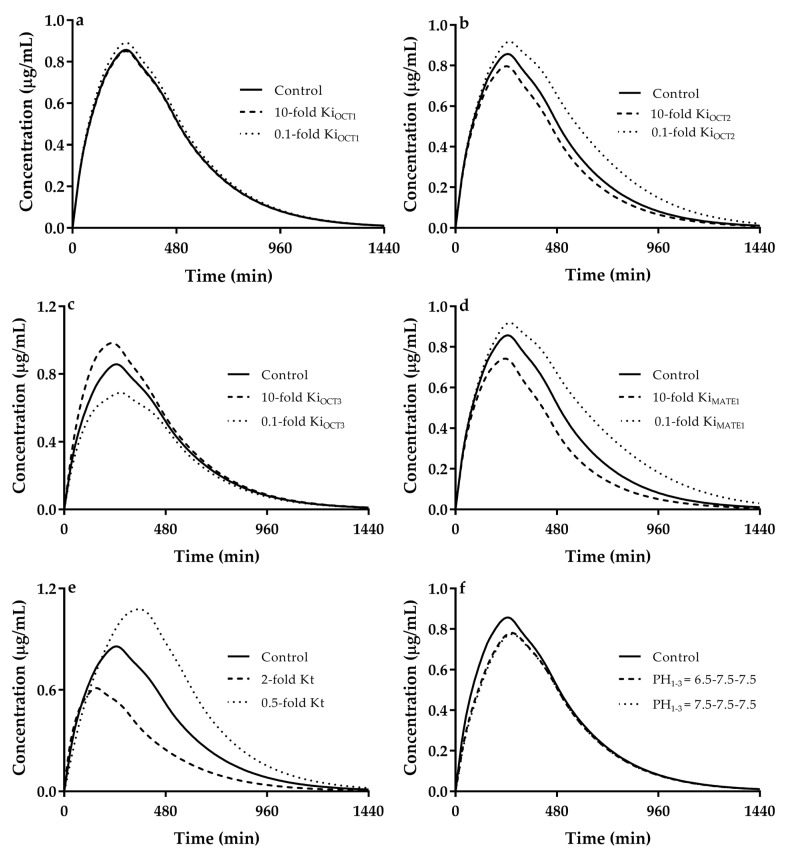
Effects of alterations in inhibition constant K_i_ values of cimetidine for (**a**) OCT1, (**b**) OCT2, (**c**) OCT3, (**d**) MATEs, (**e**) gastrointestinal transit rate constant and (**f**) intestinal pH on metformin disposition.

**Figure 6 pharmaceutics-13-00698-f006:**
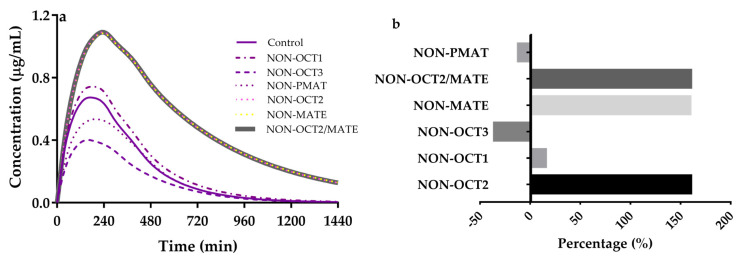
Individual contributions of (**a**) the integrated effects (CL_int,up,OCT2_ + CL_int,efflux,MATEs_, NON-OCT2/MATE), renal CL_int,efflux,MATEs_ (NON-MATE), renal CL_int,up,OCT2_ (NON-OCT2), liver CL_int,up,OCT1_ (NON-OCT1) and intestinal V_max,OCT3_ (NON-OCT3), V_max,PMAT_ (NON-PMAT) to plasma concentration of metformin; (**b**) contributions of renal CL_int,up,OCT2_, renal CL_int,efflux,MATEs_, intestinal V_max,OCT3_, intestinal V_max,PMAT_ and liver CL_int,up,OCT1_ to AUC of metformin. The percentage (%) is defined as ((AUC_non transporter_ − AUC_control_)/AUC_control_) × 100%.

**Figure 7 pharmaceutics-13-00698-f007:**
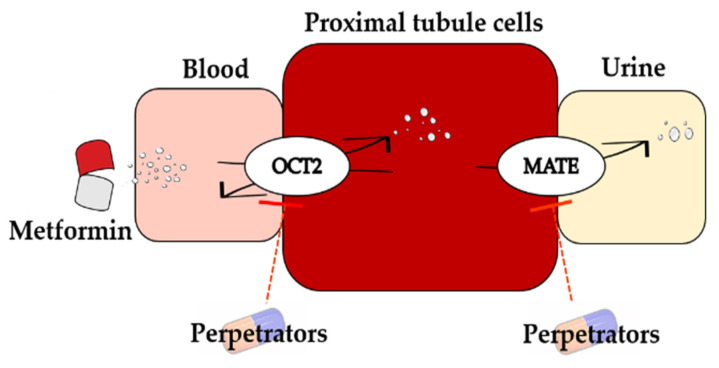
A schematic representation of transport processes for metformin and perpetrators and their interaction in renal proximal tubule cells.

**Table 1 pharmaceutics-13-00698-t001:** Pharmacokinetic parameters used for model simulation. Metformin (Met), cimetidine (Cim), pyrimethamine (Pyr), trimethoprim (Tri), ondansetron (Ond), rabeprazole (Rab) and verapamil (Ver).

Parameter	Unit	Met	Cim	Pyr	Tri	Ond	Rab	Ver
f_ub_ ^a^		1 [25]	0.82 [24]	0.15 [35]	0.43 [32]	0.325 [33]	0.04 [36]	0.14 [37]
R_b_		1 [25]	0.97 [24]	0.86 [38]	1.29 [39]	0.83 [33]	0.75 [36]	0.68 [37]
P_eff_	cm/min	0.00311 [40]	0.012 [24]	/	/	0.012 [33]		0.0156 [34]
k_a_	mL/min	/	/	0.062 [41]	0.0355 [32]		0.029 ^b^ [42]	/
CL_total_	mL/min	/	/	3.62 [43]	/	/		/
CL_int,met,h_	mL/min	110.57 [25]	188.3 [24]	27.29 [43]	52.4 [32]	1365.14 [33]	955.6 [44]	17883 [34]
CL_int,OCT1_	mL/min	63.95 [25]	200 [24]	/	/	/		/
CL_int,OCT2_	mL/min	256.45 [25]	539.8 [25]	/	/	/		/
CL_int,MATE_	mL/min	299.8 [25]	532.2 [25]	/	/	/		/
CL_renal_	mL/min	132.9	132.9	0.47 [45]	77.86 [32]	/		/
K_i,OCT1_	μM	/	101 [12]	4.46 [12]	27.7 [12]	0.27 [46]	3.0 [47]	9.62 [46]
K_i,OCT2_	μM	/	2.97 [48]	0.61 [48]	19.8 [48]	0.89 [46]	5.7 [47]	3.24 [46]
K_i,OCT3_	μM	/	45.7 [49]	>100 [48]	12.3 [48]	17.4 [49]	3.0 [47]	3.6 [46]
K_i,MATE_	μM	/	0.65 [49]	0.02 [48]	0.51 [48]	0.01 [49]	4.60 [49]	/

^a^ f_ub_ and R_b_ is unbound fraction of drug in blood and blood/plasma ratio, respectively. P_eff_ is effective permeability. CL_total_, CL_int_, and CL_renal_ are total clearance, intrinsic clearance and renal clearance, respectively. K_i_ is inhibition constant. ^b^ k_a_ of rabeprazole was simulated based on the observed data from reference.

**Table 3 pharmaceutics-13-00698-t003:** The predicted (Pre) and observed (Obs) pharmacokinetic parameters of the indicated agents as well as ratios of AUC (AUCR) or C_max_ (C_max_ R) for metformin in absence of perpetrators to that in presence of perpetrators.

Perpetrators (mg)	Drug (mg)	Ref.	C_max_ (μg/mL)		Ratio	AUC_0–t_ (μg·h/mL)		Ratio
			Pre	Obs	Pre/Obs	Pre	Obs	Pre/Obs
	Metformin (250)	[13]	0.67	0.59	1.14	5.12	4.26	1.20
Cimetidine (400)	+cimetidine		0.83	1.02	0.81	7.33	6.23	1.18
	C_max_R & AUCR		1.28	1.73	0.74	1.58	1.46	1.08
	Metformin (500)	[90]	0.95	1.19	0.80	7.93	6.58	1.21
Cimetidine (400)	+cimetidine		1.33	1.78	0.75	12.09	10.3	1.17
	C_max_R & AUCR		1.4	1.5	0.93	1.52	1.57	0.97
	Metformin (250)	[14]	0.63	0.852	0.74	4.42	3.77	1.17
Pyrimethamine (50)	+pyrimethamine		1.1	1.35	0.81	13.46	8.68	1.55
	C_max_R & AUCR		1.75	1.58	1.10	3.05	2.30	1.32
	Metformin (0.1)	[14]	0.00089	0.00042	2.12	0.0029	0.0021	1.38
Pyrimethamine (50)	+pyrimethamine		0.0012	0.0004	3.00	0.0066	0.0023	2.87
	C_max_R & AUCR		1.35	0.95	1.42	2.28	1.10	2.08
	Metformin (500)	[15]	0.93	1.14	0.82	6.95	5.91	1.18
Pyrimethamine (50)	+pyrimethamine		1.61	2.32	0.69	14.52	15.24	0.95
	C_max_R & AUCR		1.73	2.04	0.85	2.09	2.58	0.81
	Metformin (500)	[16]	1.28	1.3	0.98	8.1	6.8	1.19
Trimethoprim (200)	+trimethoprim		1.42	1.8	0.79	9.24	9.3	0.99
	C_max_R & AUCR		1.11	1.38	0.80	1.14	1.37	0.83
	Metformin (850)	[103]	1.81	1.17	1.55	16.15	6.69	2.41
Trimethoprim (200)	+trimethoprim		2.07	1.4	1.48	20.37	8.68	2.35
	C_max_R & AUCR		1.14	1.2	0.95	1.26	1.3	0.97
	Metformin (850)	[19]	1.32	2.28	0.58	12.13	15.2	0.80
Ondansetron (8)	+ondansetron		1.53	2.75	0.56	14.1	18.3	0.77
	C_max_R & AUCR		1.16	1.21	0.96	1.16	1.2	0.97
	Metformin (500)	[27]	1.42	1.1	1.29	10.24	5.9	1.74
Rabeprazole (20)	+rabeprazole		1.35	1.3	1.04	10.21	6.8	1.50
	C_max_R & AUCR		0.95	1.18	0.81	1	1.15	0.87
	Metformin (750)	[20]	1.79	4.2	0.43	13.56	24.69	0.55
Rabeprazole (20)	+rabeprazole		1.72	5	0.34	13.29	28.28	0.47
	C_max_R & AUCR		0.96	1.19	0.81	0.98	1.15	0.85
	Metformin (750)	[99]	1.22	1.51	0.81	10.35	8.22	1.26
Verapamil (180)	+verapamil		1.28	1.64	0.78	11.32	8.84	1.28
	C_max_R & AUCR		1.05	1.09	0.96	1.09	1.08	1.01

## Data Availability

All observed data including metformin and perpetrators are from reports in PubMed (https://pubmed.ncbi.nlm.nih.gov/, accessed on 13 May 2021) and listed in “References” section.

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
