# Peer review of "A Whole-Body Physiologically Based Pharmacokinetic Model Characterizing Interplay of OCTs and MATEs in Intestine, Liver and Kidney to Predict Drug-Drug Interactions of Metformin with Perpetrators"

_pharmaceutics, 2021, doi:10.3390/pharmaceutics13050698_

Round 1

Reviewer 1 Report

The study presents the development of a whole-body PBPK model characterizing interplay OCTs and MATEs in the intestine, liver and kidney to predict drug-drug interactions with metformin. The authors used multifactorial equations based on the physiological processes and pharmacokinetics of the studied drugs to build the model, which was further validated. Its parameters were undergone to sensitivity analysis.

The manuscript is well organized, all analyses seem to be appropriately performed, the conclusions are supported by the data presented.

Minor remarks:

1) Abbreviation “PMAT” should be explained in the abstract as it was done for OCTs and MATEs.

2) Please explain why you collected data from healthy volunteers and not from patients with diabetes treated with metformin and studied perpetrators.

3) Eq.(1) – explain the symbol “Kt0.

4) line 245: explain what “SERT” stands for.

5) Plots presented in Figure 4 show discrepancies between predicted and observed concentrations. As the authors explain in the discussion, the differences may be caused by several factors. Do you have any solution how to improve the developed PBPK model?

Author Response

Dear Reviewer:

Thanks for your comments, which are all valuable and very helpful for revising and improving our paper, as well as the important guiding significance to our research. We have studied comments carefully and have made correction which we hope meet with approval. Revised portion are marked in red in the paper. The main corrections in the paper and the responds to the reviewer’s comments are as flowing:

Point 1:

Abbreviation “PMAT” should be explained in the abstract as it was done for OCTs and MATEs.

Response 1:   

Thanks for your advice. We have added this in “Abstract” as follows:

In “Abstract”

“Transmembrane transport of metformin is highly controlled by transporters including organic cation transporters (OCTs), plasma membrane monoamine transporter (PMAT) and multi-drug/toxin extrusions (MATEs).”

Point 2:

Please explain why you collected data from healthy volunteers and not from patients with diabetes treated with metformin and studied perpetrators.

Response 2:   

Thanks for your advice. We are very sorry not to clearly illustrate it. Here, we aimed to investigate the DDI of metformin with perpetrators and estimate contribution of individual transporters in liver, kidney and intestine to metformin disposition. Some diseases (such as diabetes and renal failure) affected metformin pharmacokinetics via affecting physiological parameters of body and expressions/function of drug transporters. We once reported that diabetes altered metformin disposition via affecting gastrointestinal transit and renal function [1]. In order to exclude effects of disease on metformin disposition, clinical DDI data mainly come from healthy subjects.

We have rewritten the statement in “Results and Discussion” section as follows:

In “Results and Discussion”

“Some diseases (such as diabetes and renal failure) altered metformin disposition via affecting physiological parameters of body and expressions/function of drug transporters. For example, diabetes was reported to alter pharmacokinetics of metformin via affecting gastrointestinal transit and renal function [62]. To better investigate the DDI between metformin and perpetrators and to estimate contributions of drug transporters in liver, kidney and intestine to metformin pharmacokinetics, the clinical PK data mainly came from healthy volunteers, which come from clinic reports on PubMed.”

Point 3:

Eq. (1) – explain the symbol “Kt0.”

Response 3:   

Thanks for your advice. We have added this following the Ep1. as showing blow:

“Where Kt0 is constants of gastric emptying rate for stomach.”

Point 4:

line 245: explain what “SERT” stands for.

Response 4:   

Thank your advice. We rewritten “SERT” in “3.1. Collection of DDI data” as follows.

In “3.1. Collection of DDI data”

“Several evidences have demonstrated that ranks for gene expressions of transporters in human intestine [25,40,41] are OCT3 > serotonin transporter (SERT) > PMAT and OCT1.”

Point 5:

Plots presented in Figure 4 show discrepancies between predicted and observed concentrations. As the authors explain in the discussion, the differences may be caused by several factors. Do you have any solution how to improve the developed PBPK model?

Response 5:   

Thanks for your advice. We have rewritten discussion in “Results and Discussion” as following.

In “Results and Discussion”

Several different ways (such as optimizing the equations, improving the parameters, and rearranging the model structure involved in absorption, distribution, metabolism and excretion) have been tried to close the gap between prediction and observation, but the discrepancies between prediction and observation still exist (Figure. 2 and 4). The discrepancies may come from such as genotype, sex and age.

Reference

  1. Li, J.; Guo, H.F.; Liu, C.; Zhong, Z.; Liu, L.; Liu, X.D. Prediction of drug disposition in diabetic patients by means of a physiologically based pharmacokinetic model. Clin Pharmacokinet 2015, 54, 179-193, doi:10.1007/s40262-014-0192-8.

Reviewer 2 Report

The Authors developed the pharmacokinetic model concerning DDI with metformin.

Did Authors considered to validate at least one try in in vivo model ?

Why did Authors choose the drugs such as cimetidine, trimetoprim or verapamil which are now rather rarely  used?

Did Authors consider to develop the model for modified release tablets? Metformin as well as verapamil are administered also in modified release formulations.

Author Response

Dear Reviewer:

Thanks for your comments, which are all valuable and very helpful for revising and improving our paper, as well as the important guiding significance to our research. We have studied comments carefully and have made correction which we hope meet with approval. Revised portion are marked in red in the paper. The main corrections in the paper and the responds to the reviewer’s comments are as flowing:

Point 1

Did Authors considered to validate at least one try in in vivo model?

Response:   

Thanks for your advice. In fact, the predicated pharmacokinetic data and DDI data using PBPK model have been validated using clinical data (»in vivo model) reported in PubMed, although we did not design clinic trial again.

Point 2

Why did Authors choose the drugs such as cimetidine, trimethoprim or verapamil which are now rather rarely used?

Response 2:   

We are sorry not to clearly illustrate them. Here, we aimed to predict DDI of metformin with   perpetrators. It is generally accepted that metformin is typical substrate of OCTs and MATEs.  Cimetidine and trimethoprim are strong inhibitors of OCTs and MATEs, which have been recommended as clinical inhibitors for MATEs by FDA [1]. Verapamil is also inhibitor of OCTs. Moreover, clinical DDI data of metformin with these inhibitors have been widely reported.

We have rewritten the “Introduction” in the revised manuscript as follows:

In “Introduction

“These perpetrators are inhibitors of OCTs [12,17-20]. Cimetidine, pyrimethamine, trimethoprim and ondansetron are also strong inhibitors of MATE transporters [12]. Cimetidine and trimethoprim are also recommended as clinical inhibitors for MATEs by FDA [21].”

Point 3

Did Authors consider to develop the model for modified release tablets? Metformin as well as verapamil are administered also in modified release formulations.

Response 3:   

Thanks for your advice. Most of drugs we select was orally administrated to subjects in an immediate-release formulation. We aim to develop a whole-body PBPK model characterizing interplay of OCTs and MATE1/2-K in intestine, liver and kidney to predict DDIs of metformin with 6 perpetrators and investigate the contribution of individual transporters. The model for different formulation of drugs like modified release tablets is out of our consideration. But this is really a good direction to go. It may be an interesting work to develop a model to characterize the different formulation of drugs.

Reference

  1. FDA, U.S. Drug Development and Drug Interactions: Table of Substrates, Inhibitors and Inducers. Availabe online: https://www.fda.gov/drugs/drug-interactions-labeling/drug-development-and-drug-interactions-table-substrates-inhibitors-and-inducers (accessed on 03/10/2020).

Reviewer 3 Report

This study aimed to develop a whole-body PBPK model to describe the drug-drug interactions (DDIs) between metformin and several other perpetrators, in the light of interaction at the transporter level. The authors developed systems of ODEs and utilized pharmacokinetic parameter estimates from the literature to predict the concentration-time levels of metformin and compare them with literature findings. 

Major comments:

  1. The pharmacokinetic parameters were taken from the literature (Table 1). Even though these parameters may have appropriate values, it is questionable whether these values are appropriate to characterize their joint effect concomitantly, as in the case of the performed simulations.
  2. Table 2: Please provide the descriptive statistical criteria for the ratios along with criteria for the predictive performance (e.g., RSE, RMSE, etc.) 
  3. Figure 3: The shaded areas, corresponding to the 95% CIs, are too wide and overlap with each other. This comes from the high variability of the predictions. This is not a desirable characteristic of the performance.
  4. Figure 5: Visual inspection of Figure 5 reveals that even greater differences (01-10 fold) in the parameters can result in concentration differences much less compared to the prediction of the model. This finding makes the predictive ability of the model questionable. 
  5. The predicted C-t levels were compared with clinical reports. However, this indirect comparison can be influenced by many other un-controlled factors (e.g., patients' characteristics, gender, genetic profile, etc.), which make the prediction disputable. 
  6. What is the practical usefulness of such a PBPK model? Since in clinical practice, a population pharmacokinetic model can lead to the same or even better results, but being much simpler. 
    Perhaps this model is interesting in terms of explaining mechanistically the underlying mechanism, but its usefulness in practice is uncertain. 

Minor comments:

  1. Figure 5 is not cited
  2. Abstract: Please rephrase line 19, since it is confusing

Author Response

Dear Reviewer:

Thanks for your comments, which are all valuable and very helpful for revising and improving our paper, as well as the important guiding significance to our research. We have studied comments carefully and have made correction which we hope meet with approval. Revised portion are marked in red in the paper. The main corrections in the paper and the responds to the reviewer’s comments are as flowing:

Point 1

The pharmacokinetic parameters were taken from the literature (Table 1). Even though these parameters may have appropriate values, it is questionable whether these values are appropriate to characterize their joint effect concomitantly, as in the case of the performed simulations.

Response 1:   

We are sorry not to clearly illustrate it. The parameters in Table 1, describing disposition of metformin and perpetrators, were cited from the reports in PubMed. Some parameters of metformin, cimetidine, ondansetron, trimethoprim and verapamil have been validated by the PBPK in previous reports [1-6]. Absorption parameters of metformin were derived from data in Caco-2 cells. The reported Ki values of some perpetrators often showed large variations. In order to fully investigate risks of DDIs, the smallest Ki values (strongest inhibition) were selected. We have rewritten “Materials and Methods” in the revised manuscript as follows.

In “2. Materials and Methods

“Model parameters for illustrating pharmacokinetics of metformin and perpetrators as well as DDIs of metformin with perpetrators used in the PBPK model were selected according to following criterions. 1) The optimal parameters of metformin, cimetidine, ondansetron, trimethoprim and verapamil used in the PBPK model were previously reported [24-26,32-34]; 2) absorption parameter of metformin was derived from data in Caco-2 cells; 3) the reported Ki values of some perpetrators often showed large variations. In order to fully investigate risks of DDIs, the smallest Ki values (strongest inhibition) were selected. The selected model parameters were listed in Table 1.”

Point 2

Table 2: Please provide the descriptive statistical criteria for the ratios along with criteria for the predictive performance (e.g., RSE, RMSE, etc.)

Response 2:   

Thanks for your advice. According to your advice, we have added the relative squared error (RSE) and geometric mean-fold error (GMFE) in the revised manuscript in section Method” and 3.2. Quantitatively predicted disposition kinetics for metformin and perpetrators”.

In “Method”:

“Both relative squared error (RSE) (Equation. 18) and the geometric mean-fold error (GMFE) (Equation. 19) were further introduced to describe the difference between predictions and observations [60,61].

(18)

(19)

Where Prei, Obsi and  represent the predicted parameters, the observed parameters and their average values of observations, respectively. n is numbers of predictions.”

In “3.2. Quantitatively predicted disposition kinetics for metformin and perpetrators”

“RSE for Cmax and AUC were calculated to be 1.1% and 9.6%, respectively. The estimated GMFE values for Cmax and AUC were 1.004 and 0.890, near to 1.0. All these results demonstrate that both the developed PBPK model and the selected corresponding parameters are appropriate for describing pharmacokinetics of metformin and perpetrators.”

Point 3

Figure 3: The shaded areas, corresponding to the 95% CIs, are too wide and overlap with each other. This comes from the high variability of the predictions. This is not a desirable characteristic of the performance.

Response 3:   

Thanks for your advice. Visual predictive checks (VPCs) were made for validating the model in human populations and assessing the accuracy of the predictions following oral single dose (500 mg) administration of metformin to human. The confidence intervals both 95% [7,8] and 90% [9,10] have been widely used. According to your advice, the confidence intervals of 90% were documented in the study.

We have rewritten the “ 2.3. Model validation” and in Figure 3c legend in the revised manuscript.

In “ 2.3. Model validation

“Plasma concentration profiles and the plasma exposure parameters (AUC and Cmax) of metformin and 6 perpetrators following oral administration to human were first simulated using the developed PBPK model and the developed PBPK model was validated by visual predictive checks (VPCs). The 5th, 50th, and 95th percentiles of the simulations and their 90% confident intervals were plotted along with the observed data.”

In “Figure 3c”.

Figure 3. (a) Predicted plasma concentrations of metformin following different dosage of human. (b) Predicted(line) and observed(point) relationships between metformin dose and dose/AUC for different dose; shaded area, 0.5-2.0 folds of prediction; observations were cited from [13,15,16,19,20,24,64,65,80,84-109]. (c) Visual predictive checks (VPCs) of metformin plasma concentrations to time in 500 mg oral administration for human; solid line, the 50th percentiles; dashed lines, the 5 and 95th percentiles of the simulated populations; the shaded area, 90% confidence intervals of the simulated concentrations of the 5, 50 and 95th percentiles; and the point, observations, which were cited from [24,65,87,89,93,110].

Point 4

Figure 5: Visual inspection of Figure 5 reveals that even greater differences (01-10 folds) in the parameters can result in concentration differences much less compared to the prediction of the model. This finding makes the predictive ability of the model questionable.

Response 4:   

        We are sorry not to clearly illustrate it. The reported Ki values for perpetrators often showed large differences from different reports. For example, Ki values of cimetidine for OCT2. OCT3, MATE1 and MATE2-K varied 657- folds, 11-folds, 61-folds and 22-folds, respectively.0.1~10-fold variabilities for Ki were selected to sensitivity analysis. It was consistent with their contribution to metformin (MATEs»OCT2>OCT3>>OCT1) that alterations in Ki values for OCT2 and MATE1 significantly altered AUCR of metformin. For example, AUCR values at 0.1× Ki, 1×Ki (0.65 mM) and 10× Ki of MATEs were estimated to be 1.23, 1.54 and 1.98, respectively. OCT3 mainly affected metformin absorption, altering Cmax of metformin. The estimated CmaxR values at 0.1× Ki, 1×Ki (45.7 mM) and 10× Ki of OCT3 were1.47, 1.28 and 1.03, respectively. These results indicated that DDI data varied with Ki values. Alterations in Ki values for OCT1 little affected plasma exposure to metformin, which were in line with minor roles of OCT1 in metformin disposition.

We have rewritten section “3.4. Sensitivity analysis” in the revised manuscript as follows.

In “3.4. Sensitivity analysis”:

“Effects of Ki values for OCTs and MATEs, intestinal pH values and gastrointestinal transit rate on DDI of metformin were investigated using cimetidine as a representative for sensitive analysis. Ki values of cimetidine for OCT2, OCT3, MATE1 and MATE2-K [62] are reported to vary 657 folds, 11 folds, 61 folds and 22 folds, respectively. Thus, the variabilities of Ki values were set to be 0.1, 1 and 10, respectively. Variabilities in gastrointestinal transit time were set to be 0.5, 1 and 2. Moreover, function of intestinal OCT3 and PMAT is dependent on pH values. Based on physiological structural characteristics, three pH conditions were taken into account”

“The decrease in gastrointestinal transit rate by 50% of control increases AUC of metformin by 39.7%. In contrast, 2-fold increase gastrointestinal transit rate leads to decrease in AUC of metformin by 38.6%. In line with previous report [116], decrease gastrointestinal transit rates increases plasma exposure to metformin. Effects of cimetidine on plasma exposure to metformin varied with Ki values for MATEs/OCTs, which was dependent on their contribution. Alterations in Ki values for OCT2 and MATE1 significantly altered AUCR of metformin. For example, AUCR values at 0.1× Ki, 1×Ki (0.65 mM) and 10× Ki of MATEs were estimated to be 1.23, 1.54 and 1.98, respectively. OCT3 mainly affected metformin absorption, altering Cmax of metformin. The estimated CmaxR values at 0.1× Ki, 1×Ki (45.7 mM) and 10× Ki of OCT3 were 1.47, 1.28 and 1.03, respectively. Alterations in Ki values for OCT1 little affected plasma exposure to metformin, which were in line with minor roles of OCT1 in metformin disposition.”

Point 5

The predicted C-t levels were compared with clinical reports. However, this indirect comparison can be influenced by many other un-controlled factors (e.g., patients' characteristics, gender, genetic profile, etc.), which make the prediction disputable.

Response 5:   

We are sorry not to clearly illustrate it. Some factors such as patients' characteristics, gender and genetic variants may affect metformin disposition. A report showed that gender and race (Caucasians, Blacks, and Hispanics) little affected AUC of metformin [11]. It was reported that elderly subjects (68±2 years) exhibited 1.7 and 2.0 times higher average Cmax and AUC than the younger subjects (23±3 years) [12]. Subjects used in the prediction were young and healthy, which may exclude effects of both ages and disease on metformin disposition. Genetic variants of OCTs/MATEs have been demonstrated to alter metformin disposition [13,14], but results are often confused. For example, the renal clearance of metformin was reported to be unaltered in patients carrying the MATE1 variant, OCT2 and OCT3 [13]. But, volunteers with SLC22A3 variants (rs8187722 or rs2292334) were reported to have higher Cmax and AUC of metformin than the wild SLC22A3 [14]. Another report showed that compared with OCT2 reference allele (808G/G), volunteers carrying heterozygous for 808G/T had higher renal clearance and secretory clearance of metformin, accompanied by significantly lower metformin concentrations at early times after metformin administration [15]. However, c.808 (G>T) alone affected neither renal clearance nor secretory clearance of metformin, but both the renal clearance and secretory clearance were significantly increased for the volunteers with c.808 (G>T) who were also homozygous for the reference variant g.-66T>C in MATE1. On the contrast, the volunteers with c.808 (G>T) who were also heterozygous for g.-66T>C showed the lower renal clearance and secretory clearance of metformin compared with volunteers with minor alleles in c.808 (G>T) carrying the g.-66T>C reference genotype. These results indicated that c.808 (G>T) could have a dominant genotype to phenotype correlation [16].

We have rewritten the section “Result and Discussion” in the revised manuscript as follows:

In “Result and Discussion

“Several different ways (such as optimizing the equations, improving the parameters, and rearranging the model structure involved in absorption, distribution, metabolism and excretion) have been tried to close the gap between prediction and observation, but the discrepancies between prediction and observation still exist (Figure. 2 and 4). The discrepancies may come from such as genotype, sex and age. A report showed that gender and race (Caucasians, Blacks, and Hispanics) little affected AUC of metformin [120]. The elderly subjects (68±2 years) were reported to exhibit 1.7 and 2.0 times higher average Cmax and AUC than the younger subjects (23±3 years) [93]. Subjects used in the prediction were young and healthy, excluding effects of both ages and disease on metformin disposition. Genetic variants of OCTs/MATEs have been demonstrated to alter metformin disposition [10,121-123], but results are often confused. For example, the renal clearance of metformin was unaltered in patients carrying the MATE1 variant, OCT2 and OCT3 [121]. But volunteers with SLC22A3 variants (rs8187722 or rs2292334) had higher Cmax and AUC of metformin than the wild SLC22A3 [10]. Another report showed that compared with OCT2 reference allele (808G/G), volunteers carrying heterozygous for 808G/T had higher renal clearance and secretory clearance of metformin, accompanied by significantly lower metformin concentrations at early times after metformin administration [122]. However, c.808 (G>T) alone affected neither renal clearance nor secretory clearance of metformin, but both the renal clearance and secretory clearance were significantly in-creased for the volunteers with c.808 (G>T) who were also homozygous for the reference variant g.-66T>C in MATE1. On the contrast, the volunteers with c.808 (G>T) who were also heterozygous for g.-66T>C showed the lower renal clearance and secretory clearance of metformin compared with volunteers with c.808 (G>T) carrying the g.-66T>C reference genotype. These results indicated that c.808 (G>T) could have a dominant genotype to phenotype correlation [123]. Thus, contributions of OCT and MATE polymorphisms to pharmacokinetics of metformin and DDI needed further investigation.”

Point 6

What is the practical usefulness of such a PBPK model? Since in clinical practice, a population pharmacokinetic model can lead to the same or even better results but being much simpler.

Perhaps this model is interesting in terms of explaining mechanistically the underlying mechanism, but its usefulness in practice is uncertain.

Response 6:   

We are sorry not to clearly illustrate it. The PBPK modeling is considered to a powerful tool to explore and quantitatively predict the pharmacokinetics of drugs and the magnitude of DDIs. It is applied at increasingly early stages during drug development and is recommended by the US Food and Drug Administration (FDA) [17] and the European Medicines Agency (EMA) [18] for the design of clinical DDI trials and population pharmacokinetic studies.

We have rewritten “Introduction” in the revised manuscript as follows:

In “Introduction

“The physiologically based pharmacokinetic (PBPK) model is considered to a powerful tool to explore and quantitatively predict the pharmacokinetics of drugs and the magnitude of DDIs. It is widely applied at increasingly early stages during drug development and is recommended by the US Food and Drug Administration [22] and the European Medicines Agency [23] for the design of clinical DDI trials and population pharmacokinetic studies. Several investigators have successfully developed PBPK model to illustrate transporter-mediated DDI of metformin with cimetidine [24-26]. Several investigators have successfully developed PBPK model to illustrate trans-porter-mediated DDI of metformin with cimetidine [24-26]. However, these studies have focused on transporter-mediated renal secretion without considering intestinal absorption and hepatic disposition of metformin, which does not explain why some drugs (such as verapamil, trimethoprim and rabeprazole) increase plasma exposure to metformin, but little affect or even attenuate antihyperglycemic activity of metformin [15,27].

Point 7

Figure 5 is not cited.

Response 7:   

Sorry to confuse you. We have added the citation in the section “3.4. Sensitivity analysis” as following.

“It was found that the tested factors remarkably alter DDIs of metformin with cimetidine (Figure. 5).”

Point 8

Abstract: Please rephrase line 19, since it is confusing

Response 8:   

Sorry to confuse you. We have rewritten the sentence in line19 in the section “Abstract” as following.

In “Abstract”:

“Simulations showed that co-administration of perpetrators increased plasma exposures to metformin, which were consistent with clinic observations.”

Reference

  1. Nishiyama, K.; Toshimoto, K.; Lee, W.; Ishiguro, N.; Bister, B.; Sugiyama, Y. Physiologically-Based Pharmacokinetic Modeling Analysis for Quantitative Prediction of Renal Transporter-Mediated Interactions Between Metformin and Cimetidine. CPT Pharmacometrics Syst Pharmacol 2019, 8, 396-406, doi:10.1002/psp4.12398.
  2. Burt, H.J.; Neuhoff, S.; Almond, L.; Gaohua, L.; Harwood, M.D.; Jamei, M.; Rostami-Hodjegan, A.; Tucker, G.T.; Rowland-Yeo, K. Metformin and cimetidine: Physiologically based pharmacokinetic modelling to investigate transporter mediated drug-drug interactions. Eur J Pharm Sci 2016, 88, 70-82, doi:10.1016/j.ejps.2016.03.020.
  3. Hanke, N.; Turk, D.; Selzer, D.; Ishiguro, N.; Ebner, T.; Wiebe, S.; Muller, F.; Stopfer, P.; Nock, V.; Lehr, T. A Comprehensive Whole-Body Physiologically Based Pharmacokinetic Drug-Drug-Gene Interaction Model of Metformin and Cimetidine in Healthy Adults and Renally Impaired Individuals. Clin Pharmacokinet 2020, 10.1007/s40262-020-00896-w, doi:10.1007/s40262-020-00896-w.
  4. Nakada, T.; Kudo, T.; Kume, T.; Kusuhara, H.; Ito, K. Quantitative analysis of elevation of serum creatinine via renal transporter inhibition by trimethoprim in healthy subjects using physiologically-based pharmacokinetic model. Drug Metab Pharmacokinet 2018, 33, 103-110, doi:10.1016/j.dmpk.2017.11.314.
  5. Zhou, W.; Johnson, T.N.; Bui, K.H.; Cheung, S.Y.A.; Li, J.; Xu, H.; Al-Huniti, N.; Zhou, D. Predictive Performance of Physiologically Based Pharmacokinetic (PBPK) Modeling of Drugs Extensively Metabolized by Major Cytochrome P450s in Children. Clin Pharmacol Ther 2018, 104, 188-200, doi:10.1002/cpt.905.
  6. Heikkinen, A.T.; Baneyx, G.; Caruso, A.; Parrott, N. Application of PBPK modeling to predict human intestinal metabolism of CYP3A substrates - an evaluation and case study using GastroPlus. Eur J Pharm Sci 2012, 47, 375-386, doi:10.1016/j.ejps.2012.06.013.
  7. Kong, W.M.; Sun, B.B.; Wang, Z.J.; Zheng, X.K.; Zhao, K.J.; Chen, Y.; Zhang, J.X.; Liu, P.H.; Zhu, L.; Xu, R.J., et al. Physiologically based pharmacokinetic-pharmacodynamic modeling for prediction of vonoprazan pharmacokinetics and its inhibition on gastric acid secretion following intravenous/oral administration to rats, dogs and humans. Acta Pharmacol Sin 2020, 41, 852-865, doi:10.1038/s41401-019-0353-2.
  8. Chen, Y.; Zhao, K.; Liu, F.; Xie, Q.; Zhong, Z.; Miao, M.; Liu, X.; Liu, L. Prediction of Deoxypodophyllotoxin Disposition in Mouse, Rat, Monkey, and Dog by Physiologically Based Pharmacokinetic Model and the Extrapolation to Human. Front Pharmacol 2016, 7, 488, doi:10.3389/fphar.2016.00488.
  9. Chitnis, S.D.; Han, Y.; Yamaguchi, M.; Mita, S.; Zhao, R.; Sunkara, G.; Kulmatycki, K. Population pharmacokinetic modeling and noncompartmental analysis demonstrated bioequivalence between metformin component of metformin/vildagliptin fixed-dose combination products and metformin immediate-release tablet sourced from various countries. Clin Pharmacol Drug Dev 2016, 5, 40-51, doi:10.1002/cpdd.191.
  10. Yoon, H.; Cho, H.Y.; Yoo, H.D.; Kim, S.M.; Lee, Y.B. Influences of organic cation transporter polymorphisms on the population pharmacokinetics of metformin in healthy subjects. AAPS J 2013, 15, 571-580, doi:10.1208/s12248-013-9460-z.
  11. Karim, A.; Slater, M.; Bradford, D.; Schwartz, L.; Zhao, Z.; Cao, C.; Laurent, A. Oral antidiabetic drugs: bioavailability assessment of fixed-dose combination tablets of pioglitazone and metformin. Effect of body weight, gender, and race on systemic exposures of each drug. J Clin Pharmacol 2007, 47, 37-47, doi:10.1177/0091270006293755.
  12. Jang, K.; Chung, H.; Yoon, J.S.; Moon, S.J.; Yoon, S.H.; Yu, K.S.; Kim, K.; Chung, J.Y. Pharmacokinetics, Safety, and Tolerability of Metformin in Healthy Elderly Subjects. J Clin Pharmacol 2016, 56, 1104-1110, doi:10.1002/jcph.699.
  13. Tzvetkov, M.V.; Vormfelde, S.V.; Balen, D.; Meineke, I.; Schmidt, T.; Sehrt, D.; Sabolic, I.; Koepsell, H.; Brockmoller, J. The effects of genetic polymorphisms in the organic cation transporters OCT1, OCT2, and OCT3 on the renal clearance of metformin. Clin Pharmacol Ther 2009, 86, 299-306, doi:10.1038/clpt.2009.92.
  14. Hakooz, N.; Jarrar, Y.B.; Zihlif, M.; Imraish, A.; Hamed, S.; Arafat, T. Effects of the genetic variants of organic cation transporters 1 and 3 on the pharmacokinetics of metformin in Jordanians. Drug Metab Pers Ther 2017, 32, 157-162, doi:10.1515/dmpt-2017-0019.
  15. Chen, Y.; Li, S.; Brown, C.; Cheatham, S.; Castro, R.A.; Leabman, M.K.; Urban, T.J.; Chen, L.; Yee, S.W.; Choi, J.H., et al. Effect of genetic variation in the organic cation transporter 2 on the renal elimination of metformin. Pharmacogenet Genomics 2009, 19, 497-504, doi:10.1097/FPC.0b013e32832cc7e9.
  16. Christensen, M.M.; Pedersen, R.S.; Stage, T.B.; Brasch-Andersen, C.; Nielsen, F.; Damkier, P.; Beck-Nielsen, H.; Brosen, K. A gene-gene interaction between polymorphisms in the OCT2 and MATE1 genes influences the renal clearance of metformin. Pharmacogenet Genomics 2013, 23, 526-534, doi:10.1097/FPC.0b013e328364a57d.
  17. FDA, U. FDA approved drug products. Guidance for industry: physiologically based pharmacokinetic analysis. Availabe online: https://www.fda.gov/drugs/guidance-compliance-regulatory-information/guidances-drugs (accessed on 01/25/2021).
  18. (EMA), E.M.A. Guideline on the qualification and reporting of physiologically based pharmacokinetic (PBPK) modelling and simulation. Availabe online: https://www.ema.europa.eu/en/documents/scientific-guideline/draft-guideline-qualification-reporting-physiologically-based-pharmacokinetic-pbpk-modelling_en.pdf (accessed on 21 July 2016).

Reviewer 4 Report

This paper reports the PBPK modeling for prediction of transporter mediated DDI of metformin.

Overall, the development of PBPK models for metformin and each perpetrator drug and DDI modeling seem to be reasonable. Some specific points are detailed below.

  1. 2h

To better represent the modeling results of each drug, it is recommended to mark each drug with its own symbol (e.g. in a different color).

  1. 4
  • Inserted plots of concentration of metformin in liver are not observed profiles but predicted profiles. Therefore, this should be clearly presented in the figure legend and descriptions in the main text (Line-361-363).
  • It should be cautious that the predicted lower concentrations of metformin in liver are not experimental results. Please revised the part in line 361-363. The description in this section is likely to be misunderstood by the reader as a result of observation.
  1. 6a

-It seems that curves for Non-OCT2, non-MATE, and non-OCT2/MATE are overlapped completely. However, it is difficult to recognize this in the current graph form.

- The label in the figure, “NON-OCT/MATE”, should be revised as “NON-OCT2/MATE”.

  1. Section 2.2.

Some parameters in the modeling equations are not explained.

For example, partition coefficients between tissue to blood.

  1. Line 237-239

The description of supplemental table S1 and S2 has been reversed.

  1. Table 1.

The “Meta” in the first line should be “Met”. 

Author Response

Dear Reviewer:

Thanks for your comments, which are all valuable and very helpful for revising and improving our paper, as well as the important guiding significance to our research. We have studied comments carefully and have made correction which we hope meet with approval. Revised portion are marked in red in the paper. The main corrections in the paper and the responds to the reviewer’s comments are as flowing:

Point 1

    (For Figure. 2h) To better represent the modeling results of each drug, it is recommended to mark each drug with its own symbol (e.g. in a different color).

Response:   

Thanks for your advice. We have redesigned the Figure 2h as you suggest making it clear to character each drug, which is showing blow.

Figure 2. Predicted(lines)and observed (points) plasma concentrations of (a) metformin and (b) cimetidine, (c) pyrimethamine, (d) trimethoprim, (e) ondansetron, (f) rabeprazole (with the lag time because of dosage form) and (g) verapamil following oral dose to human. The observations were cited from reports [62,64-66,68,70-73,76-81]. (h) Relationship between the observed and predicted plasma concentration of the seven tested agents with different colors. Solid and dashed lines respectively represent unity and 2-fold errors between observed and predicted data.

Point 2

    (For Figure. 4) Inserted plots of concentration of metformin in liver are not observed profiles but predicted profiles. Therefore, this should be clearly presented in the figure legend and descriptions in the main text (Line-361-363).

Response 2:    

Thanks for your advice. We are sorry for unclear statement about this. We have added this description in main text (line 367-368) and in the Figure. 4 legend as following:

In main text:

“Concentrations of metformin in liver were simultaneously simulated following coadministration of these perpetrators. The simulation data shows that coadministration of trimethoprim, rabeprazole and verapamil may increase plasma concentrations of metformin, while obviously decreased concentrations of metformin. Although the decreases in concentration of metformin in liver by the three perpetrators needed to be supported by clinical data, the simulations partly explained the curious phenomenon that why co-administration of trimethoprim, rabeprazole and verapamil little affect or attenuate antihyperglycemic activity of metformin in clinical [15,99,103].”

In Figure. 4 legend

Figure 4. Predicted (line) and observed (points) plasma concentrations of the metformin alone and co-administrated with (a) cimetidine, (b) pyrimethamine, (c) trimethoprim, (d) ondansetron, (e) rabeprazole and (f) verapamil. Insert plots indicated concentrations of metformin in liver. Observations were cited from reports [13,14,16,19,24,93]. Concentrations of metformin in liver have no clinical data.”

Point 3

    It should be cautious that the predicted lower concentrations of metformin in liver are not experimental results. Please revised the part in line 361-363. The description in this section is likely to be misunderstood by the reader as a result of observation.

Response 3:   

Sorry to confuse you. We have rewritten this part as following to avoid the misunderstanding:

        “Liver is a main targeted organ for metformin. Roles of hepatic OCT1 in plasma exposure to metformin is minor, but distribution of metformin in liver is highly controlled by hepatic OCT1, in turn, affecting antihyperglycemic activity of metformin. Concentrations of metformin in liver were simultaneously simulated following coadministration of these perpetrators. The simulation data shows that coadministration of trimethoprim, rabeprazole and verapamil may increase plasma concentrations of metformin, while obviously decreased concentrations of metformin. Although the de-creases in concentration of metformin in liver by the three perpetrators needed to be supported by clinical data, the simulations partly explained the curious phenomenon that why co-administration of trimethoprim, rabeprazole and verapamil little affect or attenuate antihyperglycemic activity of metformin in clinical [15,99,103].”

Point 4

    (For 6a) It seems that curves for Non-OCT2, non-MATE, and non-OCT2/MATE are overlapped completely. However, it is difficult to recognize this in the current graph form.

Response:   

Thanks for your advice. We are sorry for unclear performance of the Figure. 6a. We have re-design the Figure.6 to make it easy to recognize as follow:

Figure 6. Individual contributions of (a) the integrated effects (CLint, up, OCT2 + CLint, efflux, MATEs, NON-OCT2/MATE), renal CLint, efflux, MATEs (NON-MATE), renal CLint, up, OCT2 (NON-OCT2), liver CLint, up, OCT1 (NON-OCT1) and intestinal Vmax, OCT3 (NON-OCT3), Vmax, PMAT (NON-PMAT) to plasma concentration of metformin; (b) contributions of renal CLint, up, OCT2, renal CLint, efflux, MATEs, intestinal Vmax, OCT3, intestinal Vmax, PMAT and liver CLint, up, OCT1 to AUC of metformin. The percentage (%) is defined as ((AUCnon transporter-AUCcontrol)/AUCcontrol) × 100%.

Point 5

    (For 6a) The label in the figure, “NON-OCT/MATE”, should be revised as “NON-OCT2/MATE”.

Response:   

Thanks for your advice. We are sorry for this kind of negligence. we have revised the label in Figure 6a.

Point 6

    Some parameters in the modeling equations are not explained.

For example, partition coefficients between tissue to blood.

Response:   

Thanks for your advice. we have added the lacking explanation of parameters mentioned in the modeling equations in “2.2. Development of PBPK model” section as following:

“Where Kti and ka,i are constants of gastric emptying rate and drug absorption rate. For example, Kt0 is constants of gastric emptying rate for stomach. Kg:b is ratio of drug concentration in gut to blood.”

“fub is unbound fraction of drug in blood.”

“Subscript “sp” indicates spleen.”

“Kr:b and Qr are ratio of drug concentration in renal to blood and blood flow in renal, respectively.”

“Ki, OCT2 and Ki, MATE is inhibition constants of perpetrators on OCT2-mediated uptake and MATE-mediated efflux.”

“Where Qtotal, Vlu, Vven, Alu and Klu:b are cardiac output, lung volume, venous volume, amount of drug in lung and ratio of drug concentration in lung to blood, respectively. Subscript “j” indicates other tissues in human body, such as heart, brain, muscle, adipose, skin, and rest tissues.”

Point 7

(For Line 237-239) The description of supplemental table S1 and S2 has been reversed.

Response:   

We are sorry for the mistake for the statement. We have corrected the description for Table S1 and S2 in “3.1. Collection of DDI data” as following:

“The ratios of drug concentration in tissue to plasma were calculated (Table S2) using method [39] based on tissue composition and physicochemical parameters of drugs. The physiological parameters (tissue blood flow and volume) (Table S1) and pharmacokinetic parameters (transporter parameters, metabolism parameters or inhibition parameter Ki) (Table 1) were cited from corresponding literatures.”

Point 8

(For Table 1) The “Meta” in the first line should be “Met”.

Response:   

Thanks for your advice. “a” is superscript in here as Meta. Because of formatting error, displayed as “Meta”. We have revised the abbreviation for metformin in Table 1 as following:

Round 2

Reviewer 3 Report

I have no further comments to make